# Accelerated discovery of highly active enzyme nanohybrids with parallelized Bayesian optimization in hybrid space

Yu Liu [1,2,5], Haoyang Hu [3,5], Yueheng Han[3], Jia Song Deon Chon[1], Chin Lee Lo[1], Zhixuan Chen[1], Zheng Zhang[1], Zhihong Yuan [3] ✉ & Jun Ge [1,4] ✉

Artificial intelligence (AI) has significantly advanced protein engineering, enabling rapid enzyme evolution for diverse applications. However, the fragile nature of biomacromolecules requires enzyme immobilization to preserve catalytic activity under harsh industrial conditions, which often restricts substrate diffusion and reduces enzymatic activity. This challenge demands extensive trial-and-error experiments to optimize immobilized carriers with high activity for different enzymes. Here we show a machine-learning-guided workflow along with an algorithm named parallelized hybrid-space Bayesian optimization (PHBO) to accelerate the discovery of nanocarriers for specific enzymes and reactions. Leveraging prior knowledge, machine learning and iterative feedback, within limited number of experiments, this workflow explores the reaction space of over $10^7$ experiments and achieves activity recovery of 100%, 90%, and 79% for glucose oxidase, catalase, and *Candida Antarctica* lipase B, respectively. These results demonstrate that data-efficient optimization can substantially accelerate the discovery of enzyme nanohybrids with high catalytic activity across diverse enzymatic systems.

From photosynthesis and nitrogen fixation to phosphorylation and carbohydrate metabolization, enzymes catalyze the transformation of energy and substance with strict precision and high efficiency[1,2]. Inspired by nature's repertoire, scientists have employed various computational and chemical methods, including directed evolution[3] and de novo design[4], to engineer enzymes with enhanced functionality or activities not observed in naturally occurring enzymes. Breakthroughs in artificial intelligence (AI) and rapid accumulation of protein structure database also facilitate more sophisticated understanding of the protein sequences-functions relationship[5], from which researchers have established more reliable libraries, more efficient high-throughput screening techniques, and more complicated generation mechanisms to explore a broader space for new enzymes. When exposed to the real industrial conditions, these biocatalysts still suffer from the fragile nature of biomacromolecules that undergo the dynamic conformational changes with high frequency and delicacy[6]. Immobilizing enzymes through bounding to a solid support represents an effective and widely used chemical approach for enzyme engineering, which can reduce the extent of free diffusion of enzymes in solution and provide a confined environment for the three-dimensional structures. This approach can improve enzyme stability in industrial applications but often leads to a serious loss of activity due to mass transfer resistance and/or synthetic conditions[7]. The diversity of enzymes and catalyzed reactions requires customized carriers to ensure activity when improving the enzyme stability and reusability[8]. However, current immobilization approaches rely on large amounts of trial-and-error experiments based on the scientific intuition from biochemists. Therefore, it is highly desirable to develop an

[1]Key Lab for Industrial Biocatalysis, Ministry of Education, Department of Chemical Engineering, Tsinghua University, Beijing, China. [2]Synthera Biotechnology Co. Ltd., Shenzhen, China. [3]The State Key Laboratory of Chemical Engineering and Low-carbon Technology, Department of Chemical Engineering, Tsinghua University, Beijing, China. [4]State Key Laboratory of Green Biomanufacturing, Beijing, China. [5]These authors contributed equally: Yu Liu, Haoyang Hu. ✉e-mail: zhihongyuan@mail.tsinghua.edu.cn; junge@mail.tsinghua.edu.cn

efficient decision-making method that can simultaneously recommend multiple synthetic parameters and accelerate the discovery of carriers which suit specific enzymes, achieving the holistic evolution of immobilized biocatalysts.

AI-driven decision-making methods have emerged as effective tools to assist experimental design, guide data collection, and interpret scientific insights in the era of intelligent synthesis[9,10]. As for the immobilized enzymes, due to the high value of enzyme molecules, the sophistication of enzymatic assays, large amount of carrier materials, and the variation of immobilizing experimental protocols, it is difficult to generate a large database of structure-activity relationship with high consistency and accuracy to train a delicate AI model. Compared with expert systems and evolution algorithms, Bayesian optimization (BO), as an efficient global optimization algorithm for expensive black-box objective functions[11,12], effectively mitigates the small sample problem by significantly enhancing sampling efficiency, and thus accelerates the discovery of new materials, including quantum dots[13], polymers[14], copolymers[15], photocatalyst[16], and photosensitizer[17]. For the intelligent synthetic condition recommendation of immobilized enzymes, two challenges remain to be addressed: hybrid variable spaces and parallelized sampling. The variety of synthetic precursors is coupled with their concentration, which generates a hybrid variable space. BO is originally designed for pure continuous variable spaces. While existing studies have extended the application of BO to pure categorical variable spaces[18,19], BO for categorical-continuous hybrid variable spaces is rarely explored. Gryffin[20] performs well in hybrid spaces but relies heavily on prior domain knowledge, as its effectiveness strongly depends on the quality and relevance of user-provided descriptors. If these descriptors are poorly correlated with the optimization objective, the advantages of Gryffin may diminish significantly. Conducting multiple sets of experiments simultaneously can significantly reduce the cost of a single trial of synthesis and evaluation, which requires multiple recommendations from the algorithm at the same time. The basic multiple acquisition function method for parallelized sampling tends to cause sample points to cluster, resulting in highly homogeneous evaluations and inefficient utilization of high-throughput experimental devices[21].

In this study, an AI-assisted workflow has been established to accelerate the evolution of immobilized enzymes with high activity targeting specific enzymes and reactions. Metal-organic frameworks (MOFs) synthesized by the coordination of metal nodes and organic ligands[22,23] were chosen as the starting point for designing the nanocarriers for enzyme immobilization. Our group[24,25] and fellow colleagues[26] pioneered the development of a one-pot co-precipitation method to immobilize enzymes by MOFs in aqueous solution inspired by biomineralization in nature, where enzyme molecules, metal ions and organic ligands were easily mixed in aqueous solution. This method is convenient and applicable to a wide range of enzymes, and the nanocomposites can maintain good stability and enzymatic activity. To discover the most suitable nanocarriers for enzyme encapsulation, 7 soluble zinc salts and 17 soluble organic ligands were screened to generate a large scope of reaction space, which on the other hand geometrically increased the number of reactions required to explore all the possibilities, far from what human researchers could endure (Supplementary Tables 1, 2, and Supplementary Note 1). Therefore, there were four crucial issues to be solved: (i) small sample problem, (ii) categorical-continuous hybrid variable space, (iii) parallelized recommendations of experimental conditions, and (iv) specificity of different enzymes. Aiming at addressing these challenges, we integrated probabilistic reparameterization, customized kernel, and the nearby liar method to propose the parallelized hybrid-space Bayesian optimization (PHBO) algorithm, accelerating the discovery of high-performance immobilized enzyme nanohybrids for specific enzymes.

## Results

### Chemical space and workflow design

Two categorical variables (zinc salt type and ligand type) and three continuous variables (zinc salt concentration, ligand concentration, and reaction time) were chosen (Fig. 1a). Seven water-soluble zinc salts and 17 water-soluble imidazolyl or similar ligands were screened and listed in Fig. 2, whose solubilities at room temperature and pressure were tested and used as the upper limits for precursor concentrations (Supplementary Tables 1 and 2). During the synthetic process, the volumes of precursor solutions were fixed, and the concentrations were varied in the limits of respective solubility. The volumes and concentrations of enzyme solutions were fixed for each type. In this context, over $10^7$ experiments were required to build an extensive database for these precursors, which was apparently impractical due to the limitation of resources and time (Supplementary Note 1). For the proposed workflow, PHBO can suggest 8 sets of experimental conditions in parallel based on the prior knowledge database, which were then artificially implemented and fed back to the model. PHBO would recommend again in these iterations, until the requirements of researchers were reached (Fig. 1b). Activity recovery (AR) was selected as the objective to optimize the overall immobilized enzymes, which contained comprehensive information during the immobilization process, including whether the reaction can form precipitates, encapsulation efficiency (EE), relative activity (RA), and separation and reuse properties (Fig. 1c). Once ideal carriers were found, detailed enzymatic properties including encapsulation efficiency and relative activity were investigated to get better understandings of the newly discovered nanohybrids.

### Parallelized hybrid-space Bayesian optimization construction

To address the challenges of modeling and optimization in hybrid categorical-continuous reaction spaces, we developed the PHBO algorithm, which integrated three key innovations: (i) a surrogate model built in a reparameterized continuous distribution space, which allowed customized GP modeling; (ii) an acquisition function defined as an expectation over categorical variable distributions, enabling efficient gradient-based optimization via Monte Carlo gradient estimation; and (iii) the nearby liar parallelized sampling method, which incrementally updated the surrogate model to diversify batch recommendations and reduce redundant sampling. A comparison of PHBO with representative hybrid-space or categorical-space BO algorithms was summarized in Table 1.

As mentioned above, the objective of the synthetic optimization was to maximize AR, that was, to identify the sample point with the highest AR within the reaction space using the fewest possible experiments:

$$(x^*, z^*) = \underset{(x,z) \in X \times Z}{\mathrm{argmax}} f(x, z) \tag{1}$$

The BO framework utilized Gaussian processes (GP) as the surrogate model of the reaction space[27,28], and PHBO followed this approach. Given the absence of an explicit ordinal relationship between the levels of categorical variables, it was necessary to customize a dedicated GP kernel for them. We first defined the kernel for categorical variables as:

$$k_{\mathrm{Cat}}(z, z') = \begin{cases} 1 & , \text{if } z = z' \\ e^{-\frac{1}{l}} & , \text{if } z \neq z' \end{cases} \tag{2}$$

For continuous variables, PHBO used the commonly adopted Matern 5/2 kernel. The complete hybrid kernel was derived from the

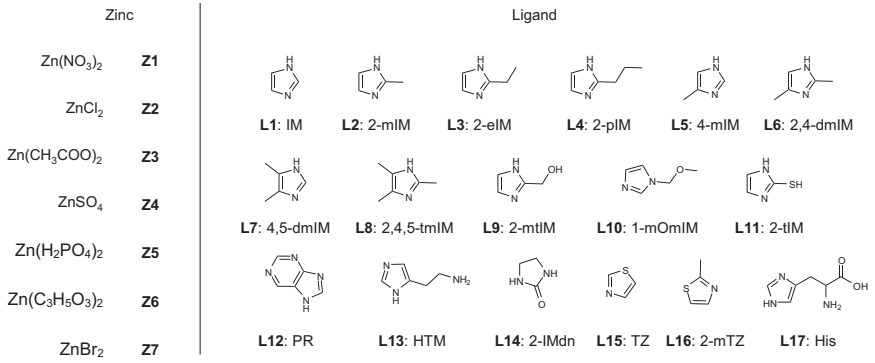

**Fig. 1 | PHBO-accelerated discovery of enzyme nanohybrids. a** One-pot co-precipitation synthesis of enzyme nanohybrids. **b** An integrated workflow using PHBO and parallel experiments to accelerate the exploration of chemical space. **c** Key parameters in the immobilization and the selection of activity recovery.

**Fig. 2 | Selection of precursors.** Zinc salts: **Z1**–**Z7**. Ligand molecules: **L1**–**L17**.

sum and product of these two kernels, weighted accordingly:

$$k_{\mathrm{Hyb}}((x,z),(x',z')) = (1-\lambda)\big(k_{\mathrm{Con}}(x,x') + k_{\mathrm{Cat}}(z,z')\big) + \lambda\big(k_{\mathrm{Con}}(x,x')k_{\mathrm{Cat}}(z,z')\big) \tag{3}$$

The kernel weights were optimized using maximum likelihood estimation, allowing data-driven adjustment of the contributions from the categorical and continuous components.

The hybrid kernel better captured the characteristics of the variables and avoided the pathological issues of the kernel for categorical variables due to the insufficient data in the early stages of iterations.

Based on the surrogate model, the BO framework iteratively constructed and optimized the acquisition function to suggest a sample point for each iteration round. Taking round t as an example:

$$(x_t, z_t) = \underset{(x,z)\in X \times Z}{\operatorname{argmax}} \ \alpha(x,z) \tag{4}$$

**Table 1 | Comparison of PHBO with existing BO algorithms for hybrid or categorical spaces**

| | COMBO[19] | Gryffin[20] | PHBO | Expected benefits of PHBO |
|---|---|---|---|---|
| Surrogate model | Graph-based GP over discrete product space using Laplacian eigenvectors | One-hot encoded GP with categorical priors | Customized GP on reparameterized distributional space | Lower complexity; less reliance on expert priors |
| Acquisition function | EI/UCB with graph enumeration search | EI/UCB with rule-based heuristics | UCB expectation over categorical variable distributions | Scalable; enables stochastic gradient optimization |
| Parallelization | Not supported | Diversified λ heuristics | Nearby liar method | Reduces sample clustering; efficient batch diversity |

To utilize efficient gradient-based optimization algorithms in the discontinuous optimization problem of the acquisition function, PHBO utilized the idea of probabilistic reparameterization. Specifically, PHBO replaced the categorical variables in the acquisition function with probability distributions (Fig. 3a, b), and accordingly replaced the acquisition function with its expectation (Fig. 3c, d):

$$(x_t, \theta_t) = \underset{(x, \theta) \in X \times \Theta}{\operatorname{argmax}} E_z[\alpha(x, z)]$$ (5)

where $z$ was drawn from the distribution $p(Z, |, \theta)$ throughout Eqs. 5–8.

The gradient of the acquisition function with respect to the continuous variables could be directly derived:

$$\frac{\partial}{\partial x} E_z[\alpha(x, z)] = \sum_{z \in Z} p(z|\theta) \frac{\partial}{\partial x} \alpha(x, z)$$ (6)

The gradient with respect to the categorical variables could be converted into an expectation form by variable substitution and then estimated through Monte Carlo sampling[29]. Specifically, we referred to the REINFORCE strategy gradient estimation method from reinforcement learning[30]:

$$\frac{\partial}{\partial \theta} E_z[\alpha(x, z)] = \sum_{z \in Z} \alpha(x, z) \frac{\partial}{\partial \theta} p(z|\theta) \approx \frac{1}{N_{MC}} \sum_{i=1}^{N_{MC}} \alpha(x, \tilde{z}_i)(I - p(\tilde{z}_i|\theta))$$ (7)

To reduce the variance of the REINFORCE estimator in (Eq. 7), we adopted a control variate technique where the acquisition function $\alpha$ was adjusted by subtracting a proportional baseline $\beta$, defined as B times the running mean of $\alpha$ from previous iterations. This yielded a modified gradient estimator (Eq. 8) with significantly lower variance. In practice, we used 128 Monte Carlo samples for each REINFORCE estimation, which represented a practical trade-off between variance reduction and computational efficiency.

$$\frac{\partial}{\partial \theta} E_z[\alpha(x, z)] \approx \frac{1}{N_{MC}} \sum_{i=1}^{N_{MC}} (\alpha(x, \tilde{z}_i) - \beta)(I - p(\tilde{z}_i|\theta))$$ (8)

With the gradients with respect to these two parts, PHBO employed the L-BFGS algorithm for acquisition function optimization to suggest the next sample point[21].

To parallelize sampling, inspired by the constant liar method[31], we developed the nearby liar method. In liar methods, parallelized recommendations were sequentially generated by adding each newly suggested sample point to the training set with a temporary "liar value", that was, a fabricated evaluation used to update the surrogate model before actual evaluation. In nearby liar, instead of assigning a fixed value, the liar value was set to the average objective value of the closest previously evaluated sample points in the reaction space, improving estimation accuracy and reducing redundant sampling. Specifically, PHBO selected a series of sample points that are closest to the suggested sample point from the evaluated dataset:

$$\operatorname{select}_{N_{NL}} \left\{ (x_{t,1i}, z_{t,1i}) | (x_{t,1i}, z_{t,1i}) \in D_t \text{ sorted by } \operatorname{Dis}((x_{t,1}, z_{t,1}), (x_{t,1i}, z_{t,1i})) \right\}$$ (9)

According to the objective function values of these points, the liar objective function value of the suggested sample point was calculated

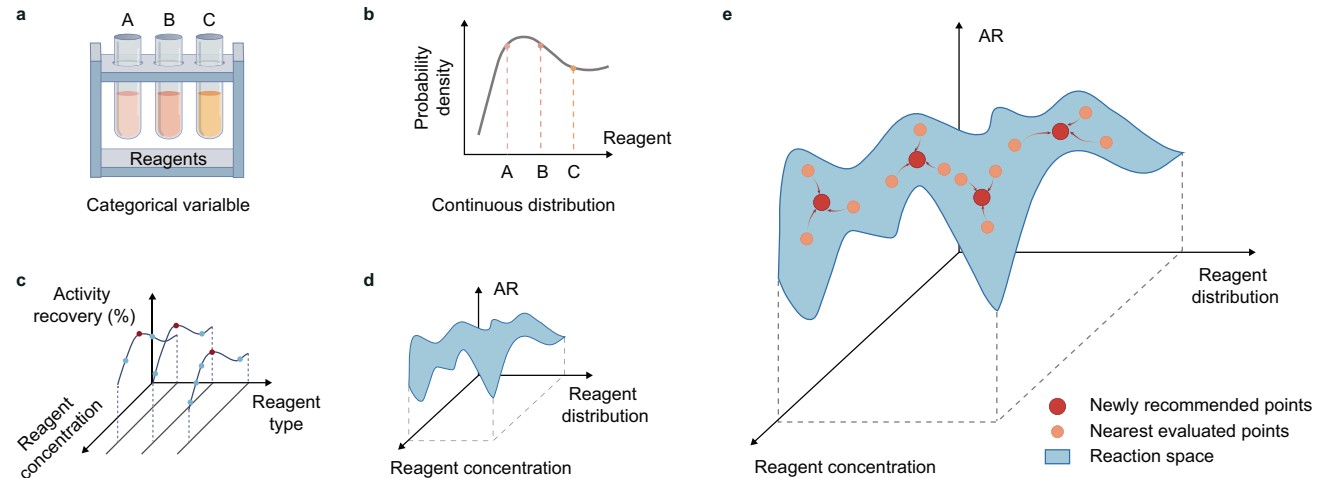

**Fig. 3 | Visualization of PHBO model. a** Discrete reagent identity (e.g., zinc salt or ligand) as a categorical variable (candidates shown were illustrative and unordered). **b** Probabilistic reparameterization of the categorical choice into a continuous reagent distribution; markers indicated the probability assigned to each reagent. **c** Discontinuous reaction space over reagent identity; red markers denoted the maximum AR for each reagent. **d** After reparameterization, a smooth objective function over distributional parameters enabled gradient-based optimization. **e** Nearby liar for parallelized recommendation: the objective of each newly recommended point (red) was temporarily imputed from several nearest evaluated points (orange) to update the surrogate model before executing the batch experiments.

and merged into the dataset (Fig. 3e):

$$D_{t,1} = D_t \cup \left\{ (x_{t,1}, z_{t,1}), \frac{\sum_{i=1}^{N_{NL}} y_{t,1i}}{N_{NL}} \right\} \quad (10)$$

Then the surrogate model was updated with the new dataset. The suggestion was repeated until the number of sample points to evaluate reaches the capacity of experimental devices. Then, batch experiments were conducted to replace all liar values with exact results, thus completing round $t$ of iterations. PHBO continued iterations until the results converged or the budget was exhausted.

The computational complexity of PHBO was primarily composed of two parts: GP model training and acquisition function optimization. The GP model training incurred a complexity of $O(n^3)$, where $n$ was the number of observed samples. The acquisition function, estimated via Monte Carlo sampling over reparametrized distributions, had an optimization complexity of $O(TMn^2)$, where $T$ was the number of optimization iterations, and $M$ was the sample size. The parallelized sampling using nearby liar introduced negligible additional overhead. Given that experimental evaluations were typically more time-consuming than these computations, PHBO offered a favorable trade-off between decision-making efficiency and computational cost.

### PHBO for the accelerated discovery of GOx nanohybrids

We first conducted ablation study of PHBO and chose glucose oxidase (GOx) as the model enzyme. One variable at a time (OVAT) by experienced chemists was employed to generate the prior knowledge database and used as a control optimization as well. By the alteration of zinc salt type, ligand type, reaction time, molar ratio of precursors, and dilution ratio of precursors, the AR was increased from 5.2% of ZIF-8 to 86.1% of Zn(2,4-dmIM)$_2$ through 47 experiments in 6 iterations (Fig. 4a and Supplementary Figs. 1, 2). Despite the nearly 17-fold increase in AR, it was difficult for OVAT to further suggest new sample points purely based on the chemist's prior experience.

LocalSearch, a BO variant that combined confidence bounds and local search[32], was a commonly used method for handling hybrid variable spaces. During the optimization of the acquisition function, LocalSearch alternated between fixing the categorical and continuous parts of the operating variables, optimizing the other part using gradient-based optimization algorithm for continuous variables and local search algorithm for categorical variables, thus suggesting the final sample point. As a comparison, the only difference between LocalSearch and PHBO was limited to the optimization of the acquisition function; and all other configurations were identical, including the nearby liar method. The AR optimized by LocalSearch exceeded OVAT in rounds 7 and 8 and reached nearly 100%, proving the effectiveness of BO framework and the nearby liar technique (Fig. 4b). PHBO, however, outperformed OVAT and LocalSearch throughout the first 6 rounds, and overtook LocalSearch in round 10 with an AR of around 110%, demonstrating the validity of probabilistic reparameterization (Fig. 4c).

The proportions of immobilized GOx with AR over 80% generated by OVAT, LocalSearch, and PHBO were 6.4%, 12.1%, and 18.0%, respectively (Fig. 4d). The reaction conditions screened by the different methods were further visualized by the precursor type, concentration, and reaction time (Fig. 4e–h and Supplementary Fig. 3). Limited by experimental experience, the chemical space explored by OVAT was regular and concentrated, rarely involving high precursor concentrations or long reaction time. LocalSearch and PHBO explored the entire variable space more evenly, with attention to long reaction time and abundant combinations of different precursors. PHBO especially focused on the low concentrations, which may produce highly active immobilized enzyme at a lower cost. Overall, PHBO was able to efficiently recommend a rich set of experimental conditions with high enzymatic activity in parallel.

The obtained reaction conditions with high activity were repeated and summarized in Supplementary Table 3. The combination of Zn(Ac)$_2$ and 2-tIM in the vicinity of a ligand/zinc molar ratio of 1.0 yield a concentrated region of high AR, which were labeled as group B, and the other combinations were labeled as group A. Scanning electron microscopy (SEM) images showed diverse morphology ranging from micron-sized lumpy precipitates to tens of nanometer-sized spherical-like particles (Fig. 4i and Supplementary Fig. 4). We discovered that enzyme nanohybrids synthesized at low ligand/zinc molar ratios

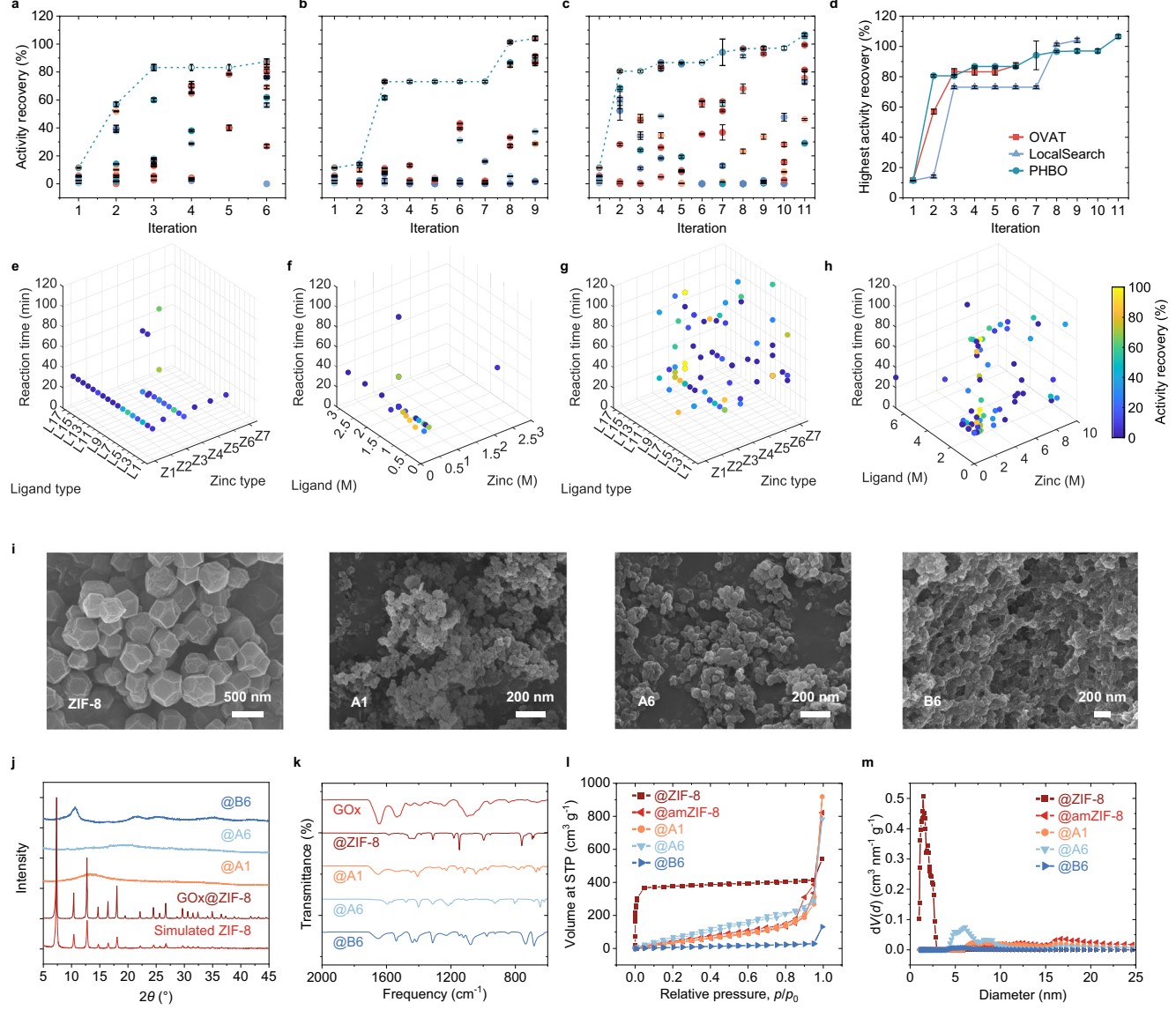

**Fig. 4 | PHBO-accelerated discovery of immobilized GOx.** Activity recovery of reactions in each iteration using **a** OVAT, **b** LocalSearch, **c** PHBO. Each point represented a single reaction, and the colors indicated the reaction index within each iteration. Data were represented as mean ± SD ($n = 3$). **d** Optimization curves of three methods. Data were represented as mean ± SD ($n = 3$). **e** Precursor types and **f** concentrations of experiments from OVAT. **g** Precursor types and **h** concentrations of experiments from PHBO. In **e–h**, each point represented a single reaction, and the colors indicated the AR of the resulting nanohybrids. **i** SEM images, **j** XRD curves, **k** IR spectra, **l** N₂ adsorption and desorption curves, and **m** pore size distribution of selected immobilized enzymes. Source data are provided as a Source data file.

presented amorphous X-ray diffraction (XRD) curves (Fig. 4j). Attenuated total reflection flourier transformed infrared spectroscopy (ATR-FTIR) confirmed the immobilization of enzymes by the appearance of the amide band (1700–1500 cm⁻¹) (Fig. 4k). The nitrogen adsorption and desorption curves showed that GOx@ZIF-8 presented a typical microporous adsorption profile with a peak at 1.43 nm and a BET surface area of 1581 m² g⁻¹ (Fig. 4l, m). Whereas, GOx@A1 and GOx@A6 presented weak mesoporous hysteresis loops that may be consistent with flaky granular particles. The pore size distribution indicated the presence of mesopores of GOx@A1 and GOx@A6, which can be attributed to particle stacking according to SEM images. BET surface areas of these two groups were 181.1 and 357.4 m² g⁻¹, respectively. The nitrogen adsorption and desorption curves and the pore size distribution of GOx@B6 indicated that there were almost no porous structures, with a BET surface area of only 50.3 m² g⁻¹.

For GOx, the optimal carrier with the highest AR was B6, which was synthesized by Zn(Ac)₂ and 2-tIM with a ligand/zinc molar ratio

of 1.4. The encapsulation efficiency and relative activity of GOx@B6 achieved nearly 100% as well. We further immobilized different types of enzymes using ZIF-8 and B6, including lactate oxidase (LOx), horseradish peroxidase (HRP), catalase (CAT), *Candida Antarctica* lipase B (CALB), and keratinase (Ker). Supplementary Fig. 5 showed that the encapsulation efficiency and relative activity of the same carrier for different enzymes varied very much, but overall, B6 outperformed ZIF-8 in five groups. B6 can achieve an encapsulation efficiency of more than 90% for five enzymes, while ZIF-8 can barely immobilize three kinds of enzymes. As for relative activity, B6 outperformed ZIF-8 except for the case of CAT. This suggested that on the one hand, B6 discovered by PHBO had significant advantages of enzyme immobilization performance compared to ZIF-8. On the other hand, the most suitable carriers for different enzymes may have different compositions and structures, which required dedicated optimization with the overall enzyme nanohybrids as the target.

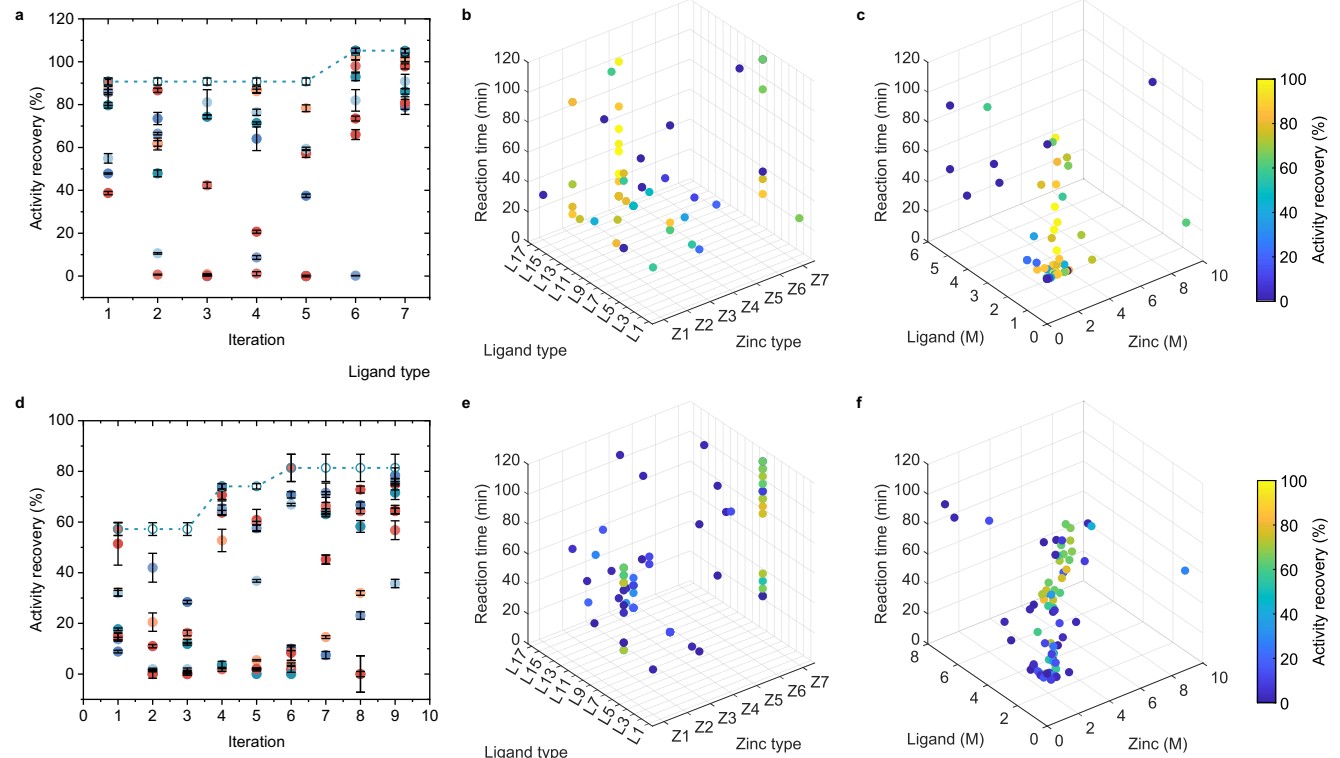

**Fig. 5 | PHBO-accelerated discovery of immobilized CAT and CALB. a** Activity recovery, **b** precursor types, and **c** concentrations of reactions in each iteration of CAT by PHBO. **d** Activity recovery, **e** precursor types, and **f** concentrations of reactions in each iteration of CALB by PHBO. In **a**, **d**, each point represented a single reaction, and the colors indicated the reaction index within each iteration. Data were represented as mean ± SD ($n = 3$). In **b**, **c**, **e**, **f**, each point represented a single reaction, and the colors indicated the AR of the resulting nanohybrids. Source data are provided as a Source data file.

## Model expansion for CAT and CALB nanohybrids

We then incorporated transfer learning into PHBO to utilize the transcendental relationship of synthetic conditions and properties from GOx to accelerate the discovery of other immobilized enzymes. The core idea of transfer learning was to extract knowledge from the source data and migrate it to the target domain to improve learning-based algorithms' performance[33–35]. Specifically, we employed two techniques: GP prior parameter update and BO warm-up. Based on the optimization process of GOx, the GP prior parameter provided a reference for the mean parameter of the GP surrogate model, while the BO warm up expanded the initial dataset of PHBO.

The first round of reaction conditions for CAT were recommended by PHBO based on the results of GOx, where the maximum of AR reached over 90%. After 6 rounds of iterations, structures with AR of 100% were discovered (Fig. 5a). Compared to 11 rounds of iterations from 11.4 to 106.6% of GOx immobilization, the validity of transfer learning was confirmed. The visualization of experimental conditions showed that PHBO explored the region of low precursor concentrations and medium reaction times to achieve high AR (Fig. 5b, c). The preferred synthetic conditions were repeated and summarized in Supplementary Table 4, some of which generated large lumpy precipitates and others small amorphous particles of tens of nanometers (Supplementary Fig. 6). Similarly, CALB was set as the target. The first round achieved an AR over 50%, compared to 1.2% of ZIF-8 as the starting point. After 10 rounds of iterations, structures with AR more than 80% were discovered (Fig. 5d). Low-to-medium precursor concentrations and long reaction times tended to produce structures with high activity as shown in Fig. 5e, f. Supplementary Table 5 showed the results of group D (ZnBr$_2$ and 2,4-dmIM) and group E (Zn(NO$_3$)$_2$ and 2,4-dmIM) with a generally broad ligand/zinc molar ratio. The morphology varied from precursor types. Group D were overall irregular

particles ranging from 100 nm to micrometers with rough surfaces, while the amorphous particles in group E were much smaller, with sizes of only 20–50 nm (Supplementary Fig. 7).

Furthermore, to quantitatively assess the effect of transfer learning, we compared PHBO to a baseline version initialized without GP prior updates nor warm-up data. For both CAT and CALB, one batch (8 parallel conditions) was recommended by the control baseline and experimentally evaluated. The results presented in Supplementary Tables 6 and 7 showed that in the first iteration, the transfer-enabled PHBO achieved average AR of 71.31% for CAT and 28.49% for CALB, while the control baseline only achieved average AR of 28.74% for CAT and 4.39% for CALB (Supplementary Notes 1 and 3). Such improvement validated the effectiveness of transfer learning and the benefit of incorporation prior knowledge from GOx optimization. These results suggested that with the assistance of transfer learning, PHBO was able to effectively utilize the prior information of existing enzymes in the database to accelerate the carriers design for other types of enzymes.

## Response surface modeling for given precursors

The optimal carriers for GOx, CAT, and CALB discovered by the PHBO-assisted workflow were different: B6: Zn(2-tIM)$_2$ (2-tIM: Zn(Ac)$_2$ = 1.4) for GOx, C2: Zn(HTM)$_2$ (HTM: Zn(Ac)$_2$ = 2.0) for CAT, and E6: Zn(2,4-dmIM)$_2$ (2,4-dmIM: Zn(NO$_3$)$_2$ = 11.1) for CALB. Combinations near the optimal molar ratio also had relatively high AR. Therefore, we conducted the response surface modeling (RSM) using the PHBO surrogate model based on the existing database to provide a thorough understanding of the selected reaction space. Figure 6a–c and Supplementary Fig. 8 presented a reference for required precursors, and highly active enzyme nanohybrids can be formed over a range of concentrations. We then selected several experimental conditions to verify the accuracy of RSM. As shown in Supplementary Table 8, the

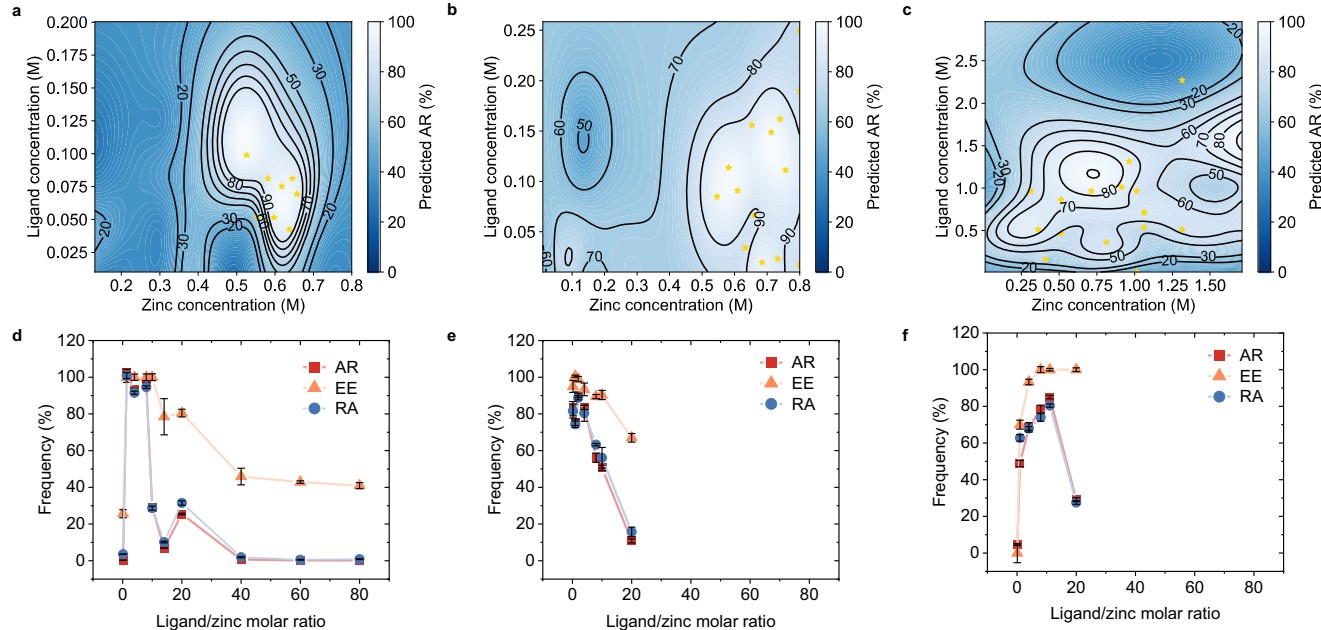

**Fig. 6 | Response surface modeling of activity recovery of three enzyme nanohybrids.** RSM images of **a** GOx ($Zn(Ac)_2$ and 2-tIM), **b** CAT ($Zn(Ac)_2$ and HTM), **c** CALB ($Zn(NO_3)_2$ and 2,4-dmIM). The predicted AR was shown as a function of zinc concentration and ligand concentration under the optimal zinc salt-ligand combination with a color map. Black contour lines denoted iso-AR levels. Yellow stars marked the experimental evaluated conditions. AR, EE, and RA of **d** GOx ($Zn(Ac)_2$ and 2-tIM), **e** CAT ($Zn(Ac)_2$ and HTM), **f** CALB ($Zn(NO_3)_2$ and 2,4-dmIM) synthesized at different molar ratios. Data were represented as mean ± SD ($n = 3$). Source data are provided as a Source data file.

PHBO predictions agreed with the general trend of the experimental results. In this way, the RSM workflow can be utilized to directly study precursor combinations of interest and provide guidance to chemical engineers in their experiments.

We then set the optimal molar ratios of three enzymes as the starting points and expanded the exploration space to study the properties of enzyme nanohybrids (Fig. 6d–f). As shown in Supplementary Figs. 9–11, the size of the particles gradually increased when increasing ligand concentrations. When the ligand/zinc molar ratio was too low, it was usually difficult to form sufficient precipitates, resulting in low encapsulation efficiency. When the molar ratio was too high, the strong alkaline environment by the hydrolysis of the ligand resulted in a more severe loss of enzyme activity, leading to low relative activity and AR. The increase of ligand concentration also decreased the formation of more particles, and CAT and CALB were unable to precipitate when the ligand/zinc molar ratio exceeded 40.

### Investigation of high activity-structure relationships

We further performed the detailed structural characterizations of the three optimal enzyme nanohybrids discovered by PHBO to uncover the origin of high activity. Take CALB as the first example. As the starting point, CALB@ZIF-8 achieved an encapsulation efficiency of 100% but suffered from very low relative activity of only 7.6%. Meanwhile, the relative activity of CALB@Zn(2,4-dmIM)$_2$ was 80.4%, resulting in a high AR of 85.1% (Fig. 7a). A scaled reaction was performed to determine the catalytic properties of both free CALB and CALB@Zn(2,4-dmIM)$_2$, where the relative activity and activity recovery showed similar results as above (Supplementary Table 9 and Supplementary Note 4).

Energy-dispersive X-ray spectroscopy (EDS) mapping confirmed the dispersed distribution of Zn and N elements from metal salts and ligands, and the distribution of S elements confirmed the immobilization of enzymes (Fig. 7b). The stretching vibration peaks of C-N at 1420–1400 cm$^{-1}$ and bending vibrational peaks at 900–650 cm$^{-1}$ of secondary amine appeared in the curve of CALB@Zn(2,4-dmIM)$_2$,

which verified the presence of 2,4-dmIM in the particles. The amide band of 1700–1500 cm$^{-1}$ in FTIR curves confirmed the immobilization of enzymes (Fig. 7c). XRD patterns verified the crystallinity of CALB@ZIF-8, while CALB@Zn(2,4-dmIM)$_2$ exhibited typical amorphous curves with broad peaks at 10°–20° (Fig. 7d). CALB@ZIF-8 possessed micropores of 0.93 nm and exhibited typical microporous adsorption curves, whereas CALB@Zn(2,4-dmIM)$_2$ had almost no porous structures and very weak adsorption capabilities (Fig. 7e). As discussed earlier in Fig. 6d–f and Supplementary Figs. 9–11, by adjusting the ligand-to-metal molar ratio under the same precursor combination, we obtained enzyme nanohybrids with different particle sizes. When the ligand-to-metal molar ratio was too low (e.g., 0.1–4.0 for CALB), it was difficult to form well-defined particles. Once stable particles were obtained, AR decreased as particle size increased. Under optimal conditions, the resulting particles exhibited negligible porosity, suggesting that enzymes were loosely confined within the solid matrix rather than encapsulated within a porous framework. In this context, smaller particles offered a higher surface area-to-volume ratio, enhancing enzyme exposure to substrates and thus contributing to the observed improvement in catalytic activity.

In addition, high-resolution Zn 2$p$ X-ray photoelectron spectroscopy (XPS) spectra revealed a shift in the Zn 2$p$ binding energy of CALB@Zn(2,4-dmIM)$_2$ toward lower energy compared to CALB@ZIF-8 (Fig. 7f), indicating a lower partial positive charge for zinc attributed to the additional methyl group of 2,4-dmIM. This would result in the relatively loose and unstable interactions between Zn and N, providing a more flexible environment to maintain enzyme structures and activity. As shown in CD diagrams, CALB released from CALB@Zn(2,4-dmIM)$_2$ showed the well-preserved secondary structures, while CALB released from CALB@ZIF-8 exhibited clear blue-shift of α-helix peaks (Fig. 7g). Moreover, amorphous particles usually suffered from instability, easy breakage, and low reusability. To observe the tolerance of particles to mechanical shear force, the immobilized enzyme suspension was treated at high centrifugation of 16,060 × $g$, 3 min, then resuspended, sampled, and tested for activity. After 10 cycles, the

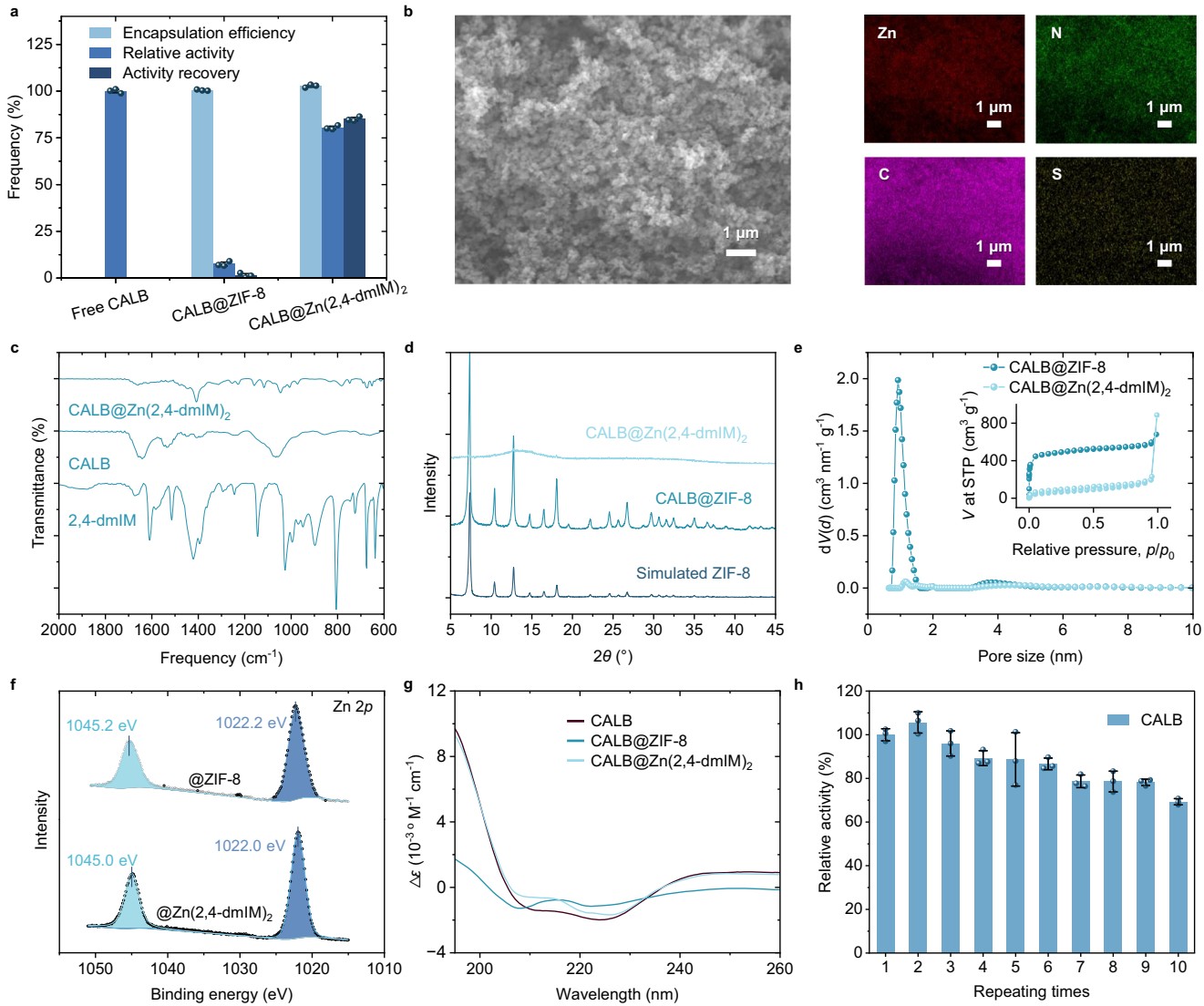

**Fig. 7 | Structural characterizations of CALB@Zn(2,4-dmIM)$_2$. a** Enzymatic activity of CALB@ZIF-8 and CALB@Zn(2,4-dmIM)$_2$. Data were represented as mean ± SD (*n* = 3). **b** SEM and EDS mapping of CALB@Zn(2,4-dmIM)$_2$. **c** FTIR curves, **d** XRD curves, **e** pore size distribution and N$_2$ adsorption and desorption curves (insert), **f** high resolution Zn 2*p* XPS spectra, and **g** CD diagram of CALB@ZIF-8 and CALB@Zn(2,4-dmIM)$_2$. **h** Relative activity of CALB@Zn(2,4-dmIM)$_2$ after being centrifugated at 16,060 × *g* for 3 min after 10 cycles. Data were represented as mean ± SD (*n* = 3). Source data are provided as a Source data file.

activity of CALB@Zn(2,4-dmIM)$_2$ remained nearly 70%, demonstrating an efficient recovery and excellent stability of the enzyme nanohybrids (Fig. 7h). GOx@Zn(2-tIM)$_2$ and CAT@Zn(HTM)$_2$ were analyzed as well, which showed similar improvement compared to GOx@ZIF-8 and CAT@ZIF-8 (Supplementary Figs. 12 and 13).

In summary, the proposed integrated workflow effectively enabled the rapid design of high-efficiency immobilization carriers for various enzymes. By loosely confining enzymes within a small solid region, the structural integrity and catalytic activity were well preserved. Leveraging transfer learning techniques, this workflow can be applied to a wide range of other enzymes for different industrial biocatalysis. Moreover, by modifying the reaction space and optimization objectives, the PHBO algorithm can facilitate the efficient design of other structures, thereby accelerating the development and iteration of functional materials.

## Discussion

A workflow incorporating parallelized Bayesian optimization in hybrid space was developed to efficiently identify high-activity immobilized enzymes within constrained experimental iterations across extensive reaction spaces. Starting from the co-precipitation synthesis of MOF, tailored immobilized carriers were engineered for GOx, CAT, and CALB, each optimized for specific catalytic functions. PHBO-generated response surface models predicted activity recovery profiles of selected precursor combinations for the scientists' reference. Structural characterization of optimal nanocarriers revealed that amorphous, nanometer-scale architectures that preserve enzyme integrity while enhancing substrate accessibility.

Beyond these primary model systems, preliminary validation on additional enzymes, including on tyrosinase (Tyr), β-galactosidase (GAL), and carbonic anhydrase (CA), further illustrated the extensibility of the workflow to broader biocatalysts without requiring major methodological adjustments (Supplementary Fig. 14 and Supplementary Note 5). Moreover, although catalytic activity was the optimization objective in this study, the AI-identified carriers also exhibited improved thermal stability relative to conventionally prepared materials on the case of CA (Supplementary Fig. 15 and Supplementary Note 6).

While the present study focused on a limited set of enzyme systems and employs artificial probes to represent catalytic performance, it served as an initial demonstration of how data-driven optimization can be integrated into enzyme immobilization design. The favorable design regions identified here could, in principle, also be discovered through conventional chemical or biochemical exploration, which on the other hand typically relied on extensive trial-and-error and became increasingly inefficient as the parameter space expands. Therefore, the primary role of the PHBO-guided workflow is to accelerate the navigation of complex, high-dimensional design spaces with substantially reduced experimental effort. Future studies may define different properties, such as long-term stability or tolerance to harsh environments, enabling the algorithm to discover carriers tailored to specific operational conditions and application needs. Overall, the algorithm's adaptability to diverse materials, reaction spaces, and optimization parameters suggest its utility as an efficient method for developing high-performance biocatalytic composites.

## Methods

### Synthesis of hybrid biocatalysts

Hybrid biocatalysts were synthesized as below. Typically, 100 μL protein solution and 160 μL zinc salt solution were simultaneously added into 1.6 mL ligand solution, and then stirred (600 rpm) at room temperature for 5 min. The precipitate was collected by centrifugation of $16,060 \times g$ for 5 min and washed 3 times with deionized water. For the synthesis of ZIF-8, 0.31 M $Zn(NO_3)_2$ and 1.25 M 2-methylimidazole were used. Concentrations of proteins were 10 mg mL$^{-1}$ for GOx, 30 mg mL$^{-1}$ for CALB, and 7.5 mg mL$^{-1}$ for CAT, respectively. Composites were freeze-dried and stored at 4 °C until further characterization.

### Characterizations of hybrid biocatalysts

Composites were sputtered with Au in a vacuum for 60 s before being observed with JOEL scanning electron microscopy (SEM) 7900 f. Samples were dispersed in ethanol and then dropped onto a carbon-coated copper grid. Transmission electron microscopy (TEM) images were taken at 200 kV by a JEOL 2100 plus equipped with an Oxford energy-dispersive X-ray (EDX) analyzer. Powder X-ray diffraction (XRD) patterns were recorded using a Bruker D8 Advance X-ray diffractometer with a 5° min$^{-1}$ scanning speed and 5°–50° diffraction angle. Circular dichroism (CD) was performed by Chirascan from Applied Photophysics Ltd. Attenuated total reflection Fourier transformed infrared spectroscopy (ATR-FTIR) was performed on Horiba Bruker Vertex 70 with ATR accessories. All characterization data were directly exported from the instrument software without additional processing.

### Enzymatic activity assays

GOx activity was measured using glucose as substrate and ABTS (2,2′-azino-bis(3-ethylbenzothiazoline-6-sulfonic acid) as the chromogenic agent. Typically, horseradish peroxidase (HRP) and ABTS were dissolved in water to make a mixed solution containing 1.5 mg mL$^{-1}$ HRP and 1.5 mg mL$^{-1}$ABTS. 30 μL of such mixed solution was added into 940 μL potassium phosphate buffer (K-PB, 10 mM, pH = 7.5) containing 50 mM glucose, and 30 μL GOx solution was added at last. The absorbance at 415 nm was recorded for 15 s.

CALB activity was measured using 4-nitrophenyl butyrate (p-NPB) as the substrate. Typically, 4 μL p-NPB was dissolved in 2 mL acetone, which was then added to 40 mL 50 mM K-PB (pH = 7.5). Fifty microliters CALB solution was added into 950 μL of such mixed solution, and the absorbance at 405 nm was recorded for 15 s.

CAT activity was measured using hydrogen peroxide as the substrate. Typically, 20 μL CAT solution was added into 980 μL potassium phosphate buffer (50 mM, pH = 7.5) containing 10 mM hydrogen peroxide. The absorbance at 240 nm was recorded for 15 s.

Tyrosinase (Tyr) activity was measured using a commercial assay kit (ACMEC, AC10628). Tyr catalyzed the oxidation of L-Dopa to dopa quinone, which subsequently reacted with MBTH to produce a chromogenic product. Reactions were conducted at 37 °C for 20 min, and the absorbance was measured at 505 nm. Enzyme activity was calculated based on the absorbance change following the manufacturer's protocol.

β-galactosidase (GAL) activity was measured using a commercial assay kit (Solarbio, BC2580). GAL hydrolyzed p-nitrophenyl-β-D-galactopyranoside (pNPGal, a standard galactosidic substrate) to yield p-nitrophenol. Reactions were conducted at 37 °C for 30 min, and the absorbance was measured at 400 nm. Enzyme activity was calculated based on the absorbance change following the manufacturer's protocol.

CA activity was measured using the Wilbur-Anderson method based on pH change. Briefly, a 20 mM TAPS buffer (solution A) and $CO_2$-saturated water (solution B) were prepared in advance. For the blank measurement, 2.45 mL of solution A and 50 μL of DI water were added to a 10 mL tube under magnetic stirring, and the time required for the pH to drop from 8.3 to 7.0 after adding solution B was recorded ($40 \pm 2$ s). For enzyme measurements, the same procedure was followed except that 50 μL of free or immobilized CA solution (diluted to 15 μg mL$^{-1}$) replaced the water. The decrease in time relative to the blank reflects catalytic activity.

For the enzymatic activity assay, each sample were tested separately for three times, and the final activity were represented as mean ± SD ($n = 3$) unless otherwise stated.

### Calculation of enzymatic properties

Initially, AR was selected as the most direct optimization target because this property contained comprehensive information of the entire immobilization process. The calculation formula was as follows:

$$AR = \frac{A_{\text{apparent activity}}}{A_{\text{total activity}}} \tag{11}$$

wherein, the total activity was the specific activity of the free enzyme times the total amount of enzyme input in the immobilization reaction.

When the iteration reached a certain stage, the immobilized material with higher AR was found, so its enzymatic properties were comprehensively measured. EE was the percentage of the amount of immobilized enzyme to the total amount of enzyme input. The amount of immobilized enzyme was obtained by subtracting the amount of enzyme in the supernatant after the reaction from the total amount of enzyme input.

$$EE = \frac{\text{Quantity}_{\text{immobilized enzymes}}}{\text{Quantity}_{\text{input enzymes}}} \tag{12}$$

RA was the proportion of residual enzyme activity in the immobilized material compared to the free enzyme at the same protein concentration. According to the encapsulation efficiency, the immobilized enzyme solution was prepared with the same concentration as the free enzyme, and the activity of both samples were tested under the same conditions. The calculation formula was as follows:

$$RA = \frac{A_{\text{apparent activity}}}{A_{\text{free enzyme activity at the sameconcentration}}} \tag{13}$$

### Reporting summary

Further information on research design is available in the Nature Portfolio Reporting Summary linked to this article.

## Data availability

The data that support the findings of this study are available from the corresponding authors upon request. Source data are provided with this paper.

## Code availability

Code used in this study, including a readme file with instructions for installing and running the code, are available from Zenodo[36] and from the corresponding authors upon request.

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

## Acknowledgements

This work was funded by the National Key Research and Development Program of China (2023YFA0913600, J.G.), National Natural Science Foundation of China (22425803, J.G.), Beijing Natural Science Foundation (Z240030, J.G.), Shenzhen Science and Technology Program (KCXFZ20240903093102004, J.G.), National Natural Science Foundation of China (21978150, Z.Y.), and the Key Research and Development Program of Shandong Province, China (2020CXGC011205, Z.Y.).

## Author contributions

Y.L., H.H. conceived the idea and designed the experiment workflow. J.G., Z.Y. supervised the project. Y.L., J.G. designed the reaction space. H.H., Y.H., and Z.Y. developed the PHBO algorithm. Y.L., J.S.D.C., C.L.L., Z.C., and Z.Z. performed the experiments and analyzed the data. Y.L., H.H., Z.Y., and J.G. wrote the manuscript.

## Competing interests

The authors declare no competing interests.

## Additional information

**Supplementary information** The online version contains Supplementary material available at https://doi.org/10.1038/s41467-026-70251-3.

