## [Transparent Peer Review File · Nature Communications]

Accelerated discovery of highly active enzyme nanohybrids with parallelized Bayesian optimization in hybrid space

Corresponding Author: Professor Jun Ge

Version 0:

Reviewer comments:

Reviewer #1

(Remarks to the Author)

The manuscript "Accelerated discovery of highly active enzyme nanohybrids with parallelized hybrid-space Bayesian optimization" by Jun Ge et al, is presenting an interesting combination of enzyme immobilization and the use of AI to efficiently sample the vast space of all possible experimental parameters to test out in order to arrive at more active catalyst. In general, the idea is excellent as there are not many studies which could address the rather "trial-error" area of enzyme immobilization where usually the through put of experimental set up is limited.

However, the manuscript is completely lacking experimental evidence which would thoroughly prove the successful outcome of the work. I do not consider spectrophotometric assays with a surrogate/chromogenic substrate as quantitative proof for assessing activity. I would expect scale up on at least few mgs of substrate, isolating and characterizing the product. Also I did not find any experimental procedure for e.g. encapsulation efficiency, calculating activity recovery and reuse performance (how was the catalyst handled between each reaction set up).

For those reasons, I can recommend the manuscript for publishing.

(Remarks on code availability)

Reviewer #2

(Remarks to the Author)

The authors present a novel PHBO method to optimize enzyme immobilization carriers based on MOFs. Their approach combines a customized Gaussian-process kernel designed for mixed categorical-continuous variable spaces, probabilistic reparameterization to enable gradient-based optimization, and a batch-suggestion strategy. The authors demonstrate significant improvement in activity recovery (AR) for immobilized enzymes compared to conventional methods. Transfer learning from GOx further reduced the optimization iterations for other enzymes. I found this work exciting, particularly because the proposed PHBO method creatively integrates multiple recent advancements in BO, such as hybrid kernel design, probabilistic reparameterization, and batch sampling. I would not hesitate to offer my support for its publication in Nat Comms and would like to suggest the following additions/corrections/etc.

1. The authors may consider more clearly differentiating their method from established methods such as COMBO (Oh et al., NeurIPS 2019) and Gryffin (by Aspuru-Guzik and co-workers). Perhaps, the authors could include a comparative table summarizing algorithmic components (e.g., kernels, recommendation strategies, etc.) alongside expected computational or performance benefits.

2. I have several queries about the following essential details:

2.1 How were kernel weights (Eq. 3) determined? Were these optimized, e.g., through maximum likelihood estimation or set manually?

2.2 In the Monte Carlo sampling (Eq. 7), what was the number of samples used for the REINFORCE gradient estimator? Were variance-reduction techniques, such as baseline subtraction, employed?

2.3 Please consider discussing or briefly reporting a computational complexity analysis.

3. The authors illustrate valuable transfer-learning capabilities from GOx to CAT and CALB, but the explanation remains qualitative. Please consider providing some quantitative metrics to evaluate transfer effectiveness.

4. The manuscript seems to only associate particle size and amorphousness with high AR. I failed to catch any discussion of

a direct causal relationship. If these two mentioned factors are the only/most significant ones, then wouldn't it make sense to conduct control experiments: synthesizing enzyme nanohybrids with consistent compositions but varied particle sizes?

5. I didn't find where "Nearby Lair" was explained or introduced.

(Remarks on code availability)

Version 1:

Reviewer comments:

Reviewer #2

(Remarks to the Author)

My comments have been fully addressed and I don't have further suggestions.

(Remarks on code availability)

Reviewer #3

(Remarks to the Author)

The authors introduce the application of AI tools to sample a set of many experimental parameters affecting the immobilization of enzymes on their activities, aiming to discover optimized experimental conditions for biocatalytic transformations. While the concept and approach are interesting, and could advance the area of biocatalysis beyond the "trial-error" method, this reviewer, being an experimentalist, rather than an AI specialist, has serious doubts regarding the overall value of the paper.

1. The selection of enzymes adapting the method is not convincing, neither are the outcoming results. The authors selected very active highly stable enzymes (almost immobilization insensitive) as test cases. Glucose oxidase, catalase and the lipase- CALB. Standard conditions given in any catalog would yield comparable activity results without any AI-predictions. If so, what are the advantages of AI?

This reviewer would like to see one or two examples of enzymes that upon immobilization under "standard" conditions lead a low activity (say 20% of the native homogeneous enzyme) and a significantly enhanced activity (60-80)% upon AI-predicted conditions. Sensitive enzymes, such as carbonic anhydrase, beta-galactosidase, formate dehydrogenase, uricase, should have been tested.

2. The enzyme assays used by the authors to probe their AI predicted results implemented non-native agents (ABTS, p-nitrophenol). Monitoring "real" biological products (CO₂ for formate dehydrogenase, bicarbonate for carbonic anhydrase or galactose for beta-galactosidase would be experimental convincing examples.

3. As pointed us by the first-round reviewers, the immobilization efficacy is not only valued by enhanced activities but, also by other environmental factors, such as temporal loss of biocatalytic activity or scaleability. This reviewer would like to see an example where 'standard" immobilization condition yield an unstable time-dependent system, whereas the AI predicted conditions lead to enhanced temporal stabilities.

(Remarks on code availability)

Version 2:

Reviewer comments:

Reviewer #3

(Remarks to the Author)

The authors made a serious effort to respond to the reviewer's comments. Nevertheless, the original doubts and critical comments are still in place, and have not been convincingly answered in the revised paper.

The authors have to admit that all the enzymes mentioned in the paper are highly stable, highly active, and can be easily immobilized. None of the enzymes reported on by the authors demonstrate the "revolutionary" AI optimized activity that could otherwise not have been achieved without the AI platform. In fact, AI-driven achievements reported by the authors could have been easily achieved by simple chemical steps available to chemists, who are active in enzyme biotechnology. The authors should concede that it is easy to kill the activity of many of the enzymes reported in the study and rescue the killing/degradation by AI recipes (but also by simple chemical steps).

Nevertheless, the concept introduced by the authors is interesting, scientifically logical and legitimate. The authors made efforts to improve the paper, and this reviewer is convinced that they believe that their attitude can serve as a powerful tool for advancing enzyme biotechnology.

Accordingly, the reviewer recommends accepting the paper. The authors, however, are advised to "tone-down" the

“revolutionary” aspect of the concept in the conclusion section, while explicitly stating the weaknesses of the present study: Implementation of highly active and stable enzymes, use of artificial synthetic non-native probes to transducing the catalytic process and lack of a unique example achievable by AI that cannot be achieved by conventional biotechnological methods.

(Remarks on code availability)

Responses to reviewers' comments and changes we have made.

Thank you very much for all the comments and suggestions from the reviewers, which have greatly improved our manuscript. Here we provided a point-to-point response and listed all new experiments and revisions in blue font. We believe that with all the efforts from our reviewers, the significance and novelty of our study has been robustly described and the revised manuscript would be suitable for publication.

Reviewer #1 (Remarks to the Author)

General Comment. *The manuscript "Accelerated discovery of highly active enzyme nanohybrids with parallelized hybrid-space Bayesian optimization" by Jun Ge et al, is presenting an interesting combination of enzyme immobilization and the use of AI to efficiently sample the vast space of all possible experimental parameters to test out in order to arrive at more active catalyst. In general, the idea is excellent as there are not many studies which could address the rather "trial-error" area of enzyme immobilization where usually the through put of experimental set up is limited.*

However, the manuscript is completely lacking experimental evidence which would thoroughly prove the successful outcome of the work. I do not consider spectrophotometric assays with a surrogate/chromogenic substrate as quantitative proof for assessing activity. I would expect scale up on at least few mgs of substrate, isolating and characterizing the product. Also I did not find any experimental procedure for e.g. encapsulation efficiency, calculating activity recovery and reuse performance (how was the catalyst handled between each reaction set up).

For those reasons, I can recommend the manuscript for publishing.

Response: We sincerely thank your appreciation of the conceptual value of combining AI-based experimental design with enzyme immobilization field. Below, we would like to address your concerns point by point:

1. The method for enzyme activity assay

The assays we used in our study are the standard method for testing enzyme activity, as can be seen in many previous studies (Nat. Commun. 2022, 13, 951., Nat. Catal. 2019, 2, 718., Nat. Commun. 2015, 6, 7240). During the activity measurement processes, the substrates we used are the real substrates for enzymes:

- For Glucose Oxidase (GOx), the activity is measured via the oxidation of the substrate-glucose, which can be further quantified through the chromogenic reaction with ABTS at 415 nm.

- Catalase (CAT) activity is measured by monitoring the decomposition of hydrogen peroxide.

- For CALB (lipase), the activity is assessed by tracking the release of p-nitrophenol from p-nitrophenyl esters which are also the standard substrates for lipase.

These methods are both reliable and provide valuable initial insights into the catalytic activity of enzymes immobilized on various carriers. We believe these assays serve as a robust measure of enzyme performance and offer the advantage of high-throughput screening in early-stage research.

2. The catalytic reactions and product isolation/characterization

Thank you for raising this point. Due to the high-throughput nature of our studies, we first focused on rapid screening of activity. Then, after finding the optimal carrier for each enzyme, we designed experiments to correlate enzyme activity with product conversion.

Taking CAT as an example, define the amount of enzyme that decomposes 1 micromole of hydrogen peroxide per minute as 1 U. The measured enzyme activity data and relative enzyme activity in a scaled-up reaction system are listed in the table. Similarly, the enzyme activity data of GOx and CALB can be calculated. According to your valuable suggestions, we have added the corresponding descriptions in the revised manuscript.

3. The encapsulation efficiency, activity recovery, and reusability

Thank you for your valuable suggestions. We have added the detailed calculation and test methods for the above properties in the revised Supporting Information. Your comments have helped us significantly improve the clarity and completeness of our manuscript.

Revisions:

Manuscript, Page 6:

Once ideal carriers were found, detailed enzymatic properties including encapsulation efficiency and relative activity were investigated to get better understandings of the newly discovered nanohybrids.

Manuscript, Page 18:

A scaled reaction was performed to determine the catalytic properties of both free CALB and CALB@Zn(2,4-dmIM)₂, where the relative activity and activity recovery showed similar results as above (Table S9).

Manuscript, Page 19:

Moreover, amorphous particles usually suffered from instability, easy breakage, and low reusability. In order to observe the tolerance of particles to mechanical shear force, the immobilized enzyme suspension was treated at high centrifugation of 13000rpm, 3 min, then resuspended, sampled, and tested for activity.

Supporting Information, Page 3:

Calculation of enzymatic properties

At the initial stage of the work, activity recovery (AR) was selected as the most direct optimization target because this property contained comprehensive information of the entire immobilization process. The calculation formula was as follows:

$$AR = \frac{A_{\text{apparent activity}}}{A_{\text{total activity}}} \times 100\%$$

wherein, the total activity was the specific activity of the free enzyme \times the total amount of enzyme input in the immobilization reaction.

After the iteration reached a certain stage, the immobilized material with higher AR was found, so its enzymatic properties were comprehensively measured. Encapsulation efficiency (EE) was the percentage of the amount of immobilized enzyme to the total amount of enzyme input. The amount of immobilized enzyme was obtained by subtracting the amount of enzyme in the supernatant after the reaction from the total amount of enzyme input.

$$EE = \frac{\text{Quantity}_{\text{immobilized enzymes}}}{\text{Quantity}_{\text{input enzymes}}} \times 100\%$$

Relative activity (RA) was the proportion of residual enzyme activity in the immobilized material compared to the free enzyme at the same protein concentration. According to the encapsulation efficiency, the immobilized enzyme solution was prepared with the same concentration as the free enzyme, and the activity of both samples were tested under the same condition. The calculation formula was as follows:

$$RA = \frac{A_{\text{apparent activity}}}{A_{\text{free enzyme activity at same } C}} \times 100\%$$

Supporting Information, Page 10:

Table S9. Comparison of prediction by PHBO and real experimental results.

	Molar extinction coefficient ($M^{-1}cm^{-1}$)	Enzyme activity (U)		Enzyme concentration in the reaction ($\mu g/mL$)	Encapsulation efficiency (%)	Relative activity (%)	Activity recovery (%)
		Free enzyme	Immobilized in best carrier				
GOx	33000 (oxidized ABTS) ^[1]	0.118±0.001	0.119±0.002	0.064	100.00	100.86	100.86
CAT	43.6 (H_2O_2) ^[2]	5.614±0.106	4.986±0.082	0.283	99.00	88.82	89.91
CALB	18000 (p-nitrophenol) ^[3]	6.150±0.130	4.946±0.068	1.475	100.00	80.43	87.70

Reaction:

GOx: 3 mL reaction system, substrate was 47 mM glucose, containing 45 $\mu g/mL$ HRP and 45 $\mu g/mL$ ABTS. The final concentration of free enzyme and immobilized enzyme was 0.064 $\mu g/mL$. The degradation of glucose was determined by the change of absorbance at 415 nm within 15s of the production of oxidized ABTS, which was the activity of GOx. The amount of enzyme that decomposed 1 micromole of glucose and generated 1 micromole of oxidized ABTS per minute was defined as 1U.

CAT: 3 mL reaction system, substrate was 9.8 mM hydrogen peroxide, the final concentration of free enzyme and immobilized enzyme was 0.283 $\mu g/mL$. The activity of CAT was determined by the decrease of absorbance at 240 nm (characteristic absorption peak of hydrogen peroxide) within 15s. The amount of enzyme that decomposed 1 micromole of hydrogen peroxide per minute was defined as 1U.

CALB: 3 mL reaction system, substrate was 0.90 $\mu g/mL$ p-NPB, the final concentration of free enzyme and immobilized enzyme was 1.475 $\mu g/mL$. The degradation of p-NPB was determined by the change of absorbance at 348 nm within 15s of the production of p-nitrophenol, which was the activity of CALB. The amount of enzyme that decomposed 1 micromole of p-NPB and generated 1 micromole of p-nitrophenol per minute was defined as 1U.

[1] Cano et al. A method to measure antioxidant activity in organic media: application to lipophilic vitamins. Redox Report, 2000, 5, 365.

[2] Nobel et al. The reaction of ferrous horseradish peroxidase with hydrogen peroxid. J. Biol. Chem. 1970, 245, 2409.

[3] Bessey et al. Preparation and measurement of the purity of the phosphatase reagent, disodium p-nitrophenyl phosphate. J. Biol. Chem. 1952, 196, 175.

Reviewer #2 (Remarks to the Author)

General Comment. *The authors present a novel PHBO method to optimize enzyme immobilization carriers based on MOFs. Their approach combines a customized Gaussian-process kernel designed for mixed categorical-continuous variable spaces, probabilistic reparameterization to enable gradient-based optimization, and a batch-suggestion strategy. The authors demonstrate significant improvement in activity recovery (AR) for immobilized enzymes compared to conventional methods. Transfer learning from GOx further reduced the optimization iterations for other enzymes. I found this work exciting, particularly because the proposed PHBO method creatively integrates multiple recent advancements in BO, such as hybrid kernel design, probabilistic reparameterization, and batch sampling. I would not hesitate to offer my support for its publication in Nat Comms and would like to suggest the following additions/corrections/etc.*

Comment 1. *The authors may consider more clearly differentiating their method from established methods such as COMBO (Oh et al., NeurIPS 2019) and Gryffin (by Aspuru-Guzik and co-workers). Perhaps, the authors could include a comparative table summarizing algorithmic components (e.g., kernels, recommendation strategies, etc.) alongside expected computational or performance benefits.*

Response: We thank the reviewer for this insightful suggestion. To better clarify the novelty and strengths of our proposed PHBO algorithm, we have added a dedicated comparison of PHBO against existing Bayesian optimization methods designed for categorical or hybrid variable spaces, namely COMBO and Gryffin. This comparison highlights key differences in three core algorithmic components: surrogate model construction, acquisition function formulation and optimization, and parallelization.

(1) Surrogate Model Construction: COMBO constructs a graph over the Cartesian product of discrete variables and defines a graph-based GP model using Laplacian eigenvectors as input features. Gryffin integrates prior expert knowledge into a weighted discrete domain using categorical priors, and then applies a one-hot encoding for GP modeling. In contrast, PHBO employs a probabilistic reparameterization strategy that maps categorical variables into a continuous distributional space. This allows the construction of a customized GP model directly in the reparameterized continuous space, reducing modeling complexity and minimizing dependence on domain-specific priors.

(2) Acquisition Function and Optimization: COMBO and Gryffin both adopt standard acquisition functions (e.g., Expected Improvement, Upper Confidence Bound). COMBO performs optimization via graph traversal or enumeration, while Gryffin uses a rule-based heuristic search over the weighted categorical space. PHBO defines the

acquisition function as the expectation over categorical distributions, which is estimated via Monte Carlo sampling. Its continuous and differentiable structure enables gradient-based optimization using stochastic methods. This formulation avoids the combinatorial explosion in COMBO when scaling to higher-dimensional categorical spaces and also mitigates Gryffin’s dependency on well-defined prior knowledge.

(3) Parallelization: COMBO does not natively support parallelization. Gryffin achieves parallelism via diversified λ -weighted acquisition functions. PHBO leverages the Nearby Liar method, which iteratively adds pseudo-evaluated points (Liar points) with estimated objective values (Liar values) derived from nearby evaluated samples. Each recommendation of PHBO is based on an updated GP model, effectively reducing sample clustering and improving batch diversity.

Revisions:

Manuscript, Page 4:

Gryffin²⁰ performs well in hybrid spaces but relies heavily on prior domain knowledge, as its effectiveness strongly depends on the quality and relevance of user-provided descriptors. If these descriptors are poorly correlated with the optimization objective, the advantages of Gryffin may diminish significantly.

Manuscript, Page 7:

To address the challenges of modeling and optimization in hybrid categorical-continuous reaction spaces, we developed the Parallelized Hybrid-space Bayesian Optimization (PHBO) algorithm. PHBO integrates three key innovations: (i) a surrogate model built in a reparametrized continuous distribution space, which allows customized GP modeling; (ii) an acquisition function defined as an expectation over categorical variable distributions, enabling efficient gradient-based optimization via Monte Carlo gradient estimation; and (iii) the Nearby Liar parallelized sampling method, which incrementally updates the surrogate model to diversify batch recommendations and reduce redundant sampling. A comparison of PHBO with representative hybrid-space or categorical-space BO algorithms is summarized in Table 1.

Table 1 Comparison of PHBO with existing BO algorithms for hybrid or categorical spaces

	COMBO ¹⁹	Gryffin ²⁰	PHBO	Expected Benefits of PHBO
Surrogate model	Graph-based GP over discrete product space using Laplacian eigenvectors	One-hot encoded GP with categorical priors	Customized GP on reparameterized distributional space	Lower complexity; less reliance on expert priors

Acquisition function	EI/UCB with graph enumeration search	EI/UCB with rule-based heuristics	UCB expectation over categorical variable distributions	Scalable; enables stochastic gradient optimization Reduces sample clustering; efficient batch diversity
Parallelization	Not supported	Diversified λ heuristics	Nearby Liar method	

Comment 2. *I have several queries about the following essential details:*

Comment 2.1 *How were kernel weights (Eq. 3) determined? Were these optimized, e.g., through maximum likelihood estimation or set manually?*

Response: We appreciate the reviewer’s question regarding the determination of kernel weights in (Eq. 3). In our implementation, the kernel weights were optimized through maximum likelihood estimation during the training of the GP surrogate model. This approach was adopted because it enables data-driven calibration of the relative contributions of the categorical and continuous components of the hybrid kernel. By maximizing the marginal likelihood of the observed data, this approach provides an efficient and principled way to tune the kernel weight, leading to improved optimization performance. This approach also avoids the need for manual tuning and allows the model to adapt to varying importance of variable types across different enzymes.

Revisions:

Manuscript, Page 8:

The kernel weights were optimized using maximum likelihood estimation, allowing data-driven adjustment of the contributions from the categorical and continuous components.

Comment 2.2 *In the Monte Carlo sampling (Eq. 7), what was the number of samples used for the REINFORCE gradient estimator? Were variance-reduction techniques, such as baseline subtraction, employed?*

Response: We are grateful to reviewer’s insightful comment. As rightly pointed out, while the REINFORCE gradient estimator provided in (Eq. 7) ((Eq. 7a) in the revised manuscript) is unbiased, its variance can be substantial, especially when the underlying distribution is broad. To mitigate this issue and enhance the computational efficiency of the PHBO, we incorporated a variance-reduction strategy based on the use of control variates.

Specifically, we employed the REINFORCE gradient estimator itself as a control variate by adjusting the score function with a proportional baseline derived from the historical mean of the acquisition function value, denoted as β . This results in the modified gradient estimator shown in Eq. (7b), where β is defined as B times the running average of α , and B is a tunable hyperparameter controlling the extent of variance reduction.

To balance accuracy and efficiency, we used 128 Monte Carlo samples per call to the REINFORCE estimator, which was found sufficient to achieve stable gradient estimation in practice.

Revisions:

Manuscript, Page 9:

We have relabeled the original Eq. (7) as Eq. (7a), and added the following paragraph to the revised manuscript: “To reduce the variance of the REINFORCE estimator in (Eq. 7a), we adopted a control variate technique where the acquisition function α is adjusted by subtracting a proportional baseline β , defined as B times the running mean of α from previous iterations. This yields a modified gradient estimator (Eq. 7b) with significantly lower variance. In practice, we used 128 Monte Carlo samples for each REINFORCE estimation, which represented a practical trade-off between variance reduction and computational efficiency.”

$$\frac{\partial}{\partial \theta} E_{z \sim p(Z|\theta)}[\alpha(x, z)] \approx \frac{1}{N_{MC}} \sum_{n=1}^{N_{MC}} (\alpha(x, \tilde{z}_i) - \beta) (I - p(\tilde{z}_i|\theta)) \quad (7b)$$

Comment 2.3 Please consider discussing or briefly reporting a computational complexity analysis.

Response: We thank the reviewer for this valuable suggestion. We have now added a brief computational complexity analysis of PHBO to better contextualize its computational cost relative to its experimental efficiency. PHBO primarily involves two computational components: (1) training of the GP surrogate model and (2) optimization of the acquisition function. The complexity of each component is as follows:

(1) GP model training. As described on Page 7 of the manuscript, PHBO employs GP as the surrogate model. The computational complexity for training the GP model is $O(n^3)$, where n is the number of observed sample points. This is consistent with the standard BO framework.

(2) Acquisition function optimization. In PHBO, acquisition function optimization is performed in the reparameterized continuous space of probability distributions. The

acquisition function is defined as an expectation over this distribution, which is estimated using Monte Carlo sampling. Let M be the number of samples per iteration, T the number of optimization iterations, the time complexity of acquisition function optimization is $O(TMn^2)$, as each Monte Carlo sample requires a GP-based prediction with complexity $O(n)$ for mean and $O(n^2)$ for variance.

To support batch recommendations, PHBO adopts the Nearby Liar strategy for parallel sampling. This method imposes minimal additional computational cost, as it leverages existing evaluations to simulate new observations and efficiently update the surrogate model.

It is important to note that in typical experimental optimization scenarios, especially in laboratory-based synthesis and evaluation, the cost of running wet experiments far exceeds the computational cost of PHBO. Therefore, PHBO remains computationally efficient and well-suited for practical applications involving expensive experimental evaluations.

Revisions:

Manuscript, Page 10:

The computational complexity of PHBO is primarily composed of two parts: GP model training and acquisition function optimization. The GP model training incurs a complexity of $O(n^3)$, where n is the number of observed samples. The acquisition function, estimated via Monte Carlo sampling over reparameterized distributions, has an optimization complexity of $O(TMn^2)$, where T is the number of optimization iterations, and M is the sample size. The parallelized sampling using Nearby Liar introduces negligible additional overhead. Given that experimental evaluations are typically more time-consuming than these computations, PHBO offers a favorable trade-off between decision-making efficiency and computational cost.

Comment 3. The authors illustrate valuable transfer-learning capabilities from GOx to CAT and CALB, but the explanation remains qualitative. Please consider providing some quantitative metrics to evaluate transfer effectiveness.

Response: We thank the reviewer for this constructive suggestion. We fully agree that assessing the effectiveness of transfer learning in a quantitative manner is important. However, due to the diversity of transfer learning strategies and the lack of a universally accepted metric to quantify source-target domain relatedness, there is currently no standardized evaluation protocol for transfer effectiveness in BO.

To address this, we adopted a controlled comparative strategy to provide an intuitive evaluation of transfer learning performance in our system. Specifically, we compared the performance of PHBO with and without transfer knowledge for both CAT and

CALB. The control setup used a “blank” PHBO initialization without GP prior parameter update nor BO warm-up, and generated one batch (8 conditions) of experimental recommendations for each enzyme. These results were then compared with the first iteration round of PHBO that included transfer knowledge from the GOx optimization.

The results showed that the transfer-enabled PHBO achieved average AR of 71.31% for CAT and 28.49% for CALB, while the control baseline only achieved average AR of 28.74% for CAT and 4.39% for CALB. Therefore, the AR achieved in the first iteration with transfer was 148% and 549% higher than that of the blank control for CAT and CALB, respectively. These quantitative comparisons clearly demonstrate the positive effect of transfer learning in accelerating the identification of high-performance enzyme nanohybrids.

Revisions:

Manuscript, Page 15:

Furthermore, to quantitatively assess the effect of transfer learning, we compared PHBO to a baseline version initialized without GP prior updates nor warm-up data. For both CAT and CALB, one batch (8 parallel conditions) was recommended by the control baseline and experimentally evaluated. The results presented in Table S6 and S7 showed that in the first iteration, the transfer-enabled PHBO achieved average AR of 51.16% for CAT and 9.77% for CALB, while the control baseline only achieved average AR of 28.74% for CAT and 4.39% for CALB. Such improvement validated the effectiveness of transfer learning and the benefit of incorporation prior knowledge from GOx optimization.

Supporting Information, Page 9-10:

Table S1. Comparison of the effectiveness of transfer learning between PHBO and control baseline: in the case of immobilized CAT biocatalysts.

Group	Zinc type	Zn conc (M)	Ligand type	Ligand conc (M)	Molar ratio of ligand/Zn	Reaction time (min)	Activity recovery (%)
PHBO	Zn(Ac) ₂	0.1550	2,4-dmIM	0.6250	40.3	30	85.7±6.1
	Zn(NO ₃) ₂	1.0616	2,4-dmIM	0.5358	5.0	30	90.8±1.6
	ZnBr ₂	1.0616	2,4-dmIM	0.5358	5.0	30	87.0±1.5
	Zn(Ac) ₂	0.6134	PR	0.1658	2.7	30	79.7±1.3
	Zn(Ac) ₂	0.2601	2-eIM	0.0100	0.4	30	54.9±2.2
	Zn(Ac) ₂	0.6333	HTM	0.0341	0.5	30	85.8±1.2
	Zn(Ac) ₂	0.5817	2-tIM	0.0814	1.4	30	47.9±0.3
	Zn(Ac) ₂	0.5817	2-tIM	0.2327	4.0	30	38.7±0.8
Control baseline	ZnSO ₄	3.9591	2-mtIM	6.2296	15.7	50	14.7±0.5
	Zn(C ₃ H ₅ O ₃) ₂	0.0039	2-IMdn	3.4604	8872.8	105	1.9±0.5

ZnSO ₄	1.9194	2,4-dmIM	8.8332	46.0	55	0.0±0.0
Zn(H ₂ PO ₄) ₂	0.2756	2-eIM	0.3678	13.3	40	54.7±2.3
Zn(Ac) ₂	0.0959	2-tIM	0.0674	7.0	80	0.6±0.6
Zn(Ac) ₂	0.6512	PR	0.5681	8.7	65	71.7±1.9
Zn(NO ₃) ₂	2.2209	2-IMdn	5.5182	24.8	40	38.1±1.3
Zn(Ac) ₂	0.6112	4-mIM	2.9172	47.7	120	48.3±0.6

Note: The control baseline experimental group was recommended without any prior knowledge. The conditions extensively explored the entire reaction space, with precursor concentrations reaching up to 8.8 M and a ligand-to-zinc ratio of 8872.8. The exploration efficiency was relatively low, resulting in an average AR of only 28.74%. In contrast, PHBO made recommendations based on knowledge extracted from the GOx experimental data. The conditions explored more of the low precursor concentration and low ligand-to-zinc ratio space, leading to an average AR of 71.31%.

Table S2. Comparison of the effectiveness of transfer learning between PHBO and control baseline: in the case of immobilized CALB biocatalysts.

Group	Zinc type	Zn conc (M)	Ligand type	Ligand conc (M)	Molar ratio of ligand/Zn	Reaction time (min)	Activity recovery (%)
PHBO	Zn(NO ₃) ₂	1.0616	2,4-dmIM	0.5358	5.0	30	51.4±8.5
	ZnBr ₂	1.0616	2,4-dmIM	0.5358	5.0	30	57.2±2.5
	Zn(Ac) ₂	0.6134	PR	0.1658	2.7	30	32.3±1.5
	Zn(Ac) ₂	0.5817	2-tIM	0.0814	1.4	30	17.7±0.5
	Zn(Ac) ₂	0.5817	2-tIM	0.2327	4.0	30	31.9±1.1
	Zn(Ac) ₂	0.6174	2-tIM	0.0752	1.2	30	13.7±0.8
	Zn(Ac) ₂	0.5817	2-tIM	0.4654	8.0	30	8.9±0.5
	Zn(Ac) ₂	0.2067	2,4-dmIM	0.8333	40.3	30	14.8±1.3
Control baseline	ZnBr ₂	8.1262	2-tIM	0.3613	0.4	50	18.3±1.7
	ZnSO ₄	0.1058	2-tIM	0.0531	5.0	85	0.0±9.0
	Zn(H ₂ PO ₄) ₂	0.3748	2-mIM	1.1350	30.3	30	1.0±0.9
	Zn(NO ₃) ₂	0.1811	PR	0.8154	45.0	100	11.2±0.7
	ZnBr ₂	9.3371	2,4-dmIM	1.6148	1.7	65	0.0±0.0
	Zn(C ₃ H ₅ O ₃) ₂	0.0214	1-mOmIM	0.525	245.3	70	0.0±0.0
	Zn(Ac) ₂	0.7879	TZ	4.9201	62.4	45	0.0±0.0
	ZnSO ₄	1.0340	4-mIM	7.6677	74.2	60	4.7±0.9

Note: Similar results can be observed for CALB as well. The control baseline experimental group explored the reaction space more widely, with a ligand-to-zinc ratio ranging from 0.4 to 245.3. CALB faced difficulty in forming nanohybrids and relatively low AR. In this case, 3 out of 8 conditions recommended by control baseline failed to form participates. The average AR generated by PHBO and control baseline were 28.49% and 4.39%, respectively.

Comment 4. *The manuscript seems to only associate particle size and amorphousness with high AR. I failed to catch any discussion of a direct causal relationship. If these two mentioned factors are the only/most significant ones, then wouldn't it make sense to conduct control experiments: synthesizing enzyme nanohybrids with consistent compositions but varied particle sizes?*

Response: We appreciate the reviewer's valuable comments. Regarding the relationship between particle size, amorphous structure, and high activity retention (AR), we have partially addressed this issue with experimental data in the manuscript and would like to further clarify as follows:

Due to the limitations of the aqueous in situ co-precipitation method, it is difficult to obtain particles with different sizes under exactly the same formulation. Therefore, we fixed the optimal precursor combination and achieved particles with varying sizes by adjusting the molar ratio of ligand to metal (Fig S9, S10, S11).

Taking CAT as an example, at lower ligand-to-metal ratios, smaller and moderately aggregated particles were formed, corresponding to higher AR. As the ratio increased, particle size gradually increased while AR decreased (Fig 4d-f). This can be explained by the fact that smaller particles provide a larger surface area, enhancing enzyme exposure and substrate diffusion, thereby improving catalytic activity; whereas excessive ligand concentration raises the reaction system's pH, which may impair enzyme activity. A similar trend was observed for GOx. As for CALB: at very low ligand-to-metal ratios (<4.0), particle morphology was irregular and immobilization efficiency was poor, yielding limited meaningful data. At moderate ratios, particle sizes were smaller, and AR reached a maximum, further supporting the above mechanistic interpretation.

In summary, our experimental results demonstrate the direct and significant impact of particle size and carrier structure on enzyme activity. Corresponding revisions have been made in the main text to clarify this point.

Revisions:

Manuscript, Page 16:

Fig. 4: Activity recovery, encapsulation efficiency, and relative activity of (d) GOx ($\text{Zn}(\text{Ac})_2$ and 2-tIM), (e) CAT ($\text{Zn}(\text{Ac})_2$ and HTM), (f) CALB ($\text{Zn}(\text{NO}_3)_2$ and 2,4-dmIM) synthesized at different molar ratios.

Manuscript, Page 18:

As discussed earlier in Fig 4d-f and S9-S11, by adjusting the ligand-to-metal molar ratio under the same precursor combination, we obtained enzyme nanohybrids with different particle sizes. When the ligand-to-metal molar ratio was too low (e.g., 0.1–4.0 for CALB), it was difficult to form well-defined particles. Once stable particles were obtained, a clear trend emerged: the activity retention (AR) decreased as particle size increased. Under optimal conditions, the resulting particles exhibited negligible porosity, suggesting that enzymes were loosely confined within the solid matrix rather than encapsulated within a porous framework. In this context, smaller particles offered a higher surface area-to-volume ratio, enhancing enzyme exposure to substrates and thus contributing to the observed improvement in catalytic activity.

Supporting Information:

Figure S9. SEM images of GOx hybrid biocatalysts synthesized with $\text{Zn}(\text{Ac})_2$ and 2-tIM with different molar ratios.

Figure S10. SEM images of CAT hybrid biocatalysts synthesized with Zn(Ac)₂ and HTM with different molar ratios.

Figure S11. SEM images of CALB hybrid biocatalysts synthesized with Zn(NO₃)₂ and 2,4-dmIM with different molar ratios.

Comment 5. *I didn't find where "Nearby Liar" was explained or introduced.*

Response: We appreciate the reviewer for pointing out this oversight. The Nearby Liar method was briefly mentioned at the bottom of Page 8 in the original manuscript (“To parallelize sampling, inspired by the Constant Liar method...”), but we agree that its mechanism was not sufficiently described.

To clarify, Nearby Liar belongs to the family of Liar methods, which are designed to enable batch acquisition in BO. These methods decompose parallelized sampling into a sequence of steps: after a sample point is suggested, it is added to the training dataset as a “Liar point” along with a “Liar value”, which is a fabricated objective value. The surrogate model is then updated based on this temporary value. This process is repeated until the desired number of sample points is collected, after which physical experiments are conducted and the Liar values are replaced with actual observations.

In contrast to the Constant Liar method, Nearby Liar improves the accuracy of the temporary evaluation by assigning the average objective value of several previously evaluated sample points that are nearest to the new sample point in the chemical space. This more accurate estimation helps reduce over-exploration and leads to more efficient parallel sampling.

Revisions:

Manuscript, Page 9:

We have modified the last paragraph in this page in the revised manuscript: “To parallelize sampling, inspired by the Constant Liar method³¹, we developed the Nearby

Liar method. In Liar methods, parallelized recommendations are sequentially generated by adding each newly suggested sample point to the training set with a temporary “Liar value”, that is, a fabricated evaluation used to update the surrogate model before actual evaluation. In Nearby Liar, instead of assigning a fixed value, the Liar value is set to the average objective value of the closest previously evaluated sample points in the reaction space, improving estimation accuracy and reducing redundant sampling. after suggesting a sample point, PHBO does not immediately conduct the experiment but uses the Nearby Lair method to continue suggesting new sample points. Specifically, PHBO selects a series of sample points that are closest to the suggested sample point from the evaluated dataset:”

Responses to reviewers' comments and changes we have made.

Thank you very much for all the comments and suggestions from the editor and reviewers, which have greatly improved our manuscript. Here we provided a point-to-point response and listed all new experiments and revisions in blue font. We believe that with all the efforts from our editor and reviewers, the significance and novelty of our study has been robustly described and the revised manuscript would be suitable for publication.

Reviewer #2 (Remarks to the Author)

My comments have been fully addressed and I don't have further suggestions.

Response: We sincerely thank you for recognizing and supporting our first round of revisions. Our manuscript has been significantly improved due to your review.

Reviewer #3 (Remarks to the Author)

Comment 1: The authors introduce the application of AI tools to sample a set of many experimental parameters affecting the immobilization of enzymes on their activities, aiming to discover optimized experimental conditions for biocatalytic transformations. While the concept and approach are interesting, and could advance the area of biocatalysis beyond the "trial-error" method, this reviewer, being an experimentalist, rather than an AI specialist, has serious doubts regarding the overall value of the paper.

1. The selection of enzymes adapting the method is not convincing, neither are the outcoming results. The authors selected very active highly stable enzymes (almost immobilization insensitive) as test cases. Glucose oxidase, catalase and the lipase-CALB. Standard conditions given in any catalog would yield comparable activity results without any AI-predictions. If so, what are the advantages of AI?

This reviewer would like to see one or two examples of enzymes that upon immobilization under "standard" conditions lead a low activity (say 20% of the native homogeneous enzyme) and a significantly enhanced activity (60-80)% upon AI-predicted conditions. Sensitive enzymes, such as carbonic anhydrase, beta-galactosidase, formate dehydrogenase, uricase, should have been tested.

Response: We thank the reviewer for this comment. Regarding the use of more sensitive enzymes, we would like to respectfully clarify that GOx, CALB, and CAT should not be regarded as "insensitive" enzymes for immobilization. Many publications have demonstrated that conventional immobilization methods and commonly used

carrier materials frequently lead to low relative activities or limited encapsulation efficiencies for these enzymes. For example, in *Biosensors & Bioelectronics* 6 (1991) 663, GOx immobilized on a glutaraldehyde-crosslinked Pt electrode retained only approximately 24.6% of the native enzyme activity (24,036 vs. 97,822 units). In *Nature Communications* 13 (2022) 951, even when using ZIF-8 as the carrier, different crystallization pathways yielded markedly divergent outcomes, with GOx@BZIF-8-S retaining only 3-5% of the free-enzyme activity. In *Biochemical Engineering Journal* 112 (2016) 20, GOx immobilized on multi-walled carbon nanotubes exhibited a relative activity of merely ~20%. For CALB, *Journal of Nanomaterials* (2021) 8812240 reported that immobilization on silica nanoparticles with varying surface areas and hydrophobicity produced relative activity retention ranging widely from 12.5% to 100%. Likewise, in *Langmuir* 40 (2024) 16338, CAT immobilized on unoptimized plastic nanobeads retained only ~22.2% activity.

Consistent with these prior observations, our own measurements using ZIF-8, one of the most widely adopted MOF carriers, resulted in low relative activities for all three enzymes (27.9%, 62.3%, and 7.6% for GOx, CAT, and CALB, respectively, data shown in Figure S5). These results showed that these three enzymes used in our work still suffered from substantial activity loss under many conventional immobilization conditions. Therefore, further exploration was required to identify the most suitable immobilization carriers for each enzyme.

Another reason for selecting GOx, CALB, and CAT was that these enzymes were among the most extensively used ones in industrial biocatalysis, with well-established enzymatic assays. These features make them ideal benchmarks for evaluating whether a new AI-based optimization method is accurate, reproducible, and generally applicable. Starting from low relative activity obtained with baseline ZIF-8, our AI-guided workflow was able to identify optimized conditions that increased the relative activity to 100%, 88.8%, and 80.4%, respectively (data shown in Table S9). This significant improvement proved the effectiveness and the general scope of the method.

In addition, we fully acknowledge the reviewer's interest in more fragile enzymes. To address this point, we have conducted supplementary experiments on tyrosinase (Tyr), β -galactosidase (GAL), and carbonic anhydrase (CA). In these cases, we performed a small number of iterative rounds (three cycles) to identify the most promising carriers for each enzyme. Only the initial and final results were reported here, as the intermediate formulations are part of ongoing studies regarding the applications of these enzymes to be published separately. Even with limited optimization, we observed substantial improvements in both encapsulation efficiency and relative activity

compared with the initial ZIF-8 baseline. Specifically, the relative activity of Tyr, GAL, and CA immobilized with ZIF-8 were 19.29%, 0.16%, and -0.03%, respectively, while those immobilized with their respective optimized carrier increased to 77.57%, 47.06%, and 88.82%, respectively (Figure S14). These preliminary results further support the reliability and extensibility of this AI-guided workflow across a broader range of enzymes. At the same time, more comprehensive optimization and application-oriented studies on these enzymes are currently ongoing in our laboratory and will be reported in future publications, as they extend beyond the scope of the present methodological study.

Revisions:

Manuscript, Page 20:

Beyond these primary model systems, preliminary validation on additional enzymes including on tyrosinase (Tyr), β -galactosidase (GAL), and carbonic anhydrase (CA) further illustrated the extensibility of the workflow to broader biocatalysts without requiring major methodological adjustments (Fig S14).

Supporting Information, Page 27:

Figure S14. (a) Encapsulation efficiency and (b) relative activity of different enzymes immobilized by ZIF-8 and respective best carriers. Data were represented as mean \pm SD (n=3).

Respective best carrier: For each enzyme, PHBO was initiated from ZIF-8 and iteratively screened 8 carriers per round for three rounds. The carrier yielding the highest activity recovery was selected as the “best carrier” and its encapsulation efficiency and relative activity were subsequently measured. Even with limited optimization, substantial improvements were observed in both encapsulation efficiency and relative activity compared with the initial ZIF-8 baseline. Specifically, the relative activity of Tyr, GAL, and CA immobilized with ZIF-8 were 19.29%, 0.16%, and -0.03%,

respectively, while those immobilized with their respective best carrier increased to 77.57%, 47.06%, and 88.82%, respectively.

2. *The enzyme assays used by the authors to probe their AI predicted results implemented non-native agents (ABTS, p-nitrophenol). Monitoring “real” biological products (CO₂ for formate dehydrogenase, bicarbonate for carbonic anhydrase or galactose for beta-galactosidase would be experimental convincing examples.*

Response: We thank the reviewer for this comment. The three enzymes evaluated in our original submission (GOx, CAT, and CAL) were evaluated using their native or native-like substrates. Specifically:

- GOx activity was measured using glucose (its natural substrate), with ABTS serving only as a chromogenic probe for the H₂O₂ generated during the native oxidation reaction.
- CAT activity was directly measured from the decomposition rate of its natural substrate, H₂O₂.
- CALB (lipase) activity was measured using p-nitrophenyl esters (p-NPB), which were widely accepted native-like substrates that mimic natural ester hydrolysis for lipase activity evaluation in enzyme immobilization studies.

As we noted in our response to Reviewer #1 in the last revision, the activity assays used in this study were standard and widely adopted methods in the enzyme immobilization literature (Nat. Commun. 2022, 13, 951., Nat. Catal. 2019, 2, 718., Nat. Commun. 2015, 6, 7240). Because our work aimed to establish a new AI-assisted optimization workflow which required screening hundreds of immobilization conditions across a high-dimensional parameter space, it was essential to employ activity assays that were high-throughput, reproducible, and readily standardized. In this context, colorimetric assays provide a reliable and literature-validated proxy for preliminary activity evaluation.

After identifying the optimal carrier materials for each enzyme, we further conducted more detailed analysis including morphology, relative activity, and encapsulation efficiency. Additionally, we evaluated the catalytic performance toward each enzyme's native substrate. These results have been provided in the Supporting Information Table S9.

Similarly, the other three enzymes (Tyr, GAL, and CA) were evaluated likewise based on their native or physiologically relevant substrates. Specifically:

- Tyr activity was measured using its natural substrate L-Dopa, which was enzymatically oxidized to dopa-quinone and subsequently reacted with MBTH to

form a chromogenic product detectable at 505 nm.

- GAL activity was measured using p-nitrophenyl- β -D-galactopyranoside (pNPGal), a chromogenic analog of lactose that retained the same β -1,4-galactosidic linkage that could be hydrolyzed by β -GAL. pNPGal was a standard galactosidic substrate whose cleavage yield p-nitrophenol with a characteristic absorbance at 400 nm.
- CA activity was measured using its native substrate CO₂ by Wilbur-Anderson assay, which was a classical and widely adopted method for measuring CA activity. In this case, the accelerated pH drop reflected the rapid formation of bicarbonate and the associated release of protons, thereby directly reporting catalytic turnover toward the natural substrate.

Revisions:

Supporting Information, Page 3:

Tyrosinase (Tyr) activity was measured using a commercial assay kit (ACMEC, AC10628). Tyr catalyzed the oxidation of L-Dopa to dopa quinone, which subsequently reacted with MBTH to produce fuchsia substance with absorbance at 505 nm. Reactions were performed according to the manufacturer's instructions.

β -Galactosidase (GAL) activity was measured using a commercial assay kit (Solarbio, BC2580). GAL hydrolyzed p-nitrophenyl- β -D-galactopyranoside (pNPGal, a standard galactosidic substrate) to yield p-nitrophenol, with a characteristic absorbance at 400 nm. Reactions were performed according to the manufacturer's instructions.

CA activity was measured using the Wilbur-Anderson method based on pH change. Briefly, a 20 mM TAPS buffer (solution A) and CO₂-saturated water (solution B) were prepared in advance. For the blank measurement, 2.45 mL of solution A and 50 μ L of DI water were added to a 10 mL tube under magnetic stirring, and the time required for the pH to drop from 8.3 to 7.0 after adding solution B was recorded (40 ± 2 s). For enzyme measurements, the same procedure was followed except that 50 μ L of free or immobilized CA solution (diluted to 15 μ g/mL) replaced the water. The decrease in time relative to the blank reflects catalytic activity.

3. As pointed us by the first-round reviewers, the immobilization efficacy is not only valued by enhanced activities but, also by other environmental factors, such as temporal loss of biocatalytic activity or scalability. This reviewer would like to see an example where 'standard' immobilization condition yield an unstable time-dependent system, whereas the AI predicted conditions lead to enhanced temporal stabilities.

Response: We thank the reviewer for this comment. We acknowledge the reviewer's concern on the temporal and environmental stability. In this study, our optimization objective was enzyme activity, and therefore the AI-derived conditions were not explicitly directed toward maximizing stability. Nevertheless, we conducted a preliminary experiment using carbonic anhydrase (CA), a representative enzyme that required tolerance to high-temperature and strongly alkaline amine solvents in industrial chemical absorption process. N-methyldiethanolamine (MDEA) was a widely used organic amine solvent in chemical absorption of CO₂-capture process, whose adsorption mechanism could synergize well with CA, as the bicarbonate generated by the enzyme was a key intermediate in MDEA-based carbon capture (Chemical Engineering Journal 2017, 307, 776). Therefore, we evaluated CA in 30% MDEA solution at 80 °C, which was a deliberately defined condition that better reflected real industrial process.

Because the ZIF-8 formulation used in our workflow did not preserve CA activity well (Figure S14), we adopted a literature-reported ZIF-L synthesis with higher baseline activity as the “standard immobilization” control (Separation and Purification Technology 2023, 315, 123683). Under the heated and alkaline condition, the standard immobilized CA rapidly lost activity, whereas the AI-identified best carrier maintained 32.3% activity after 6 h of heating. These results showed that, even though stability was not part of the optimization target, the AI-guided carrier nevertheless conferred enhanced resistance to harsh conditions.

We fully agree that stability is crucial for industrial biocatalysis. Importantly, such parameters can be directly set as optimization objectives in the PHBO workflow. This provides a clear path for future work, where long-term stability or tolerance to harsh environments can be directly set as the optimization function to find the most suitable carriers to different conditions and demands.

Revisions:

Manuscript, Page 20:

Moreover, although catalytic activity was the optimization objective in this study, the AI-identified carriers also exhibited improved stability relative to conventionally prepared materials on the case of CA (Fig S15). In the future work, different properties could be directly defined as the optimization objectives, for example, long-term stability or tolerance to harsh environments, enabling the algorithm to discover carriers tailored to specific operational conditions and application needs.

Supporting Information, Page 28:

Figure S15. Thermal stability of CA immobilized on different carriers in 30% MDEA solution at 80 °C. N-methyldiethanolamine (MDEA) was a widely used organic amine solvent in chemical absorption of CO₂-capture process.

Relative activity was monitored over time to compare the stability of CA immobilized by AI-identified carriers and control formulations. The ZIF-L carrier was chosen as the control^[1] and synthesized following the procedure reported in the reference. Under the heated and alkaline condition, CA@ZIF-L rapidly lost activity, whereas the AI-identified best carrier maintained 32.3% activity after 6 h of heating. Data were represented as mean ± SD (n=3).

[1] Peijing Shao, et al. Shape controlled ZIF-8 crystals for carbonic anhydrase immobilization to boost CO₂ uptake into aqueous MDEA solution. Separation and Purification Technology, Volume 315, 2023, 123683.

Responses to reviewers' comments and changes we have made.

Thank you very much for all the comments and suggestions from the editor and reviewers, which have greatly improved our manuscript. Here we provided a point-to-point response and listed all revisions in blue font. We believe that with all the efforts from our editor and reviewers, the significance and novelty of our study has been robustly described, and the revised manuscript would be suitable for publication.

Reviewer #3 (Remarks to the Author)

Comments:

The authors made a serious effort to respond to the reviewer's comments. Nevertheless, the original doubts and critical comments are still in place, and have not been convincingly answered in the revised paper.

The authors have to admit that all the enzymes mentioned in the paper are highly stable, highly active, and can be easily immobilized. None of the enzymes reported on by the authors demonstrate the "revolutionary" AI optimized activity that could otherwise not have been achieved without the AI platform. In fact, AI-driven achievements reported by the authors could have been easily achieved by simple chemical steps available to chemists, who are active in enzyme biotechnology. The authors should concede that it is easy to kill the activity of many of the enzymes reported in the study and rescue the killing/degradation by AI recipes (but also by simple chemical steps).

Nevertheless, the concept introduced by the authors is interesting, scientifically logical and legitimate. The authors made efforts to improve the paper, and this reviewer is convinced that they believe that their attitude can serve as a powerful tool for advancing enzyme biotechnology.

Accordingly, the reviewer recommends accepting the paper. The authors, however, are advised to "tone-down" the "revolutionary" aspect of the concept in the conclusion section, while explicitly stating the weaknesses of the present study: Implementation of highly active and stable enzymes, use of artificial synthetic non-native probes to transducing the catalytic process and lack of a unique example achievable by AI that cannot be achieved by conventional biotechnological methods.

Response:

We thank the reviewer for the appreciation of the conceptual interest and scientific soundness of the study. In the meantime, we have revised the Discussion part according to the reviewer's suggestion to better present the scope of the work in a more balanced

manner. We now explicitly state the current limitations of the study, including enzyme selection, probe design, and the absence of AI-exclusive examples. We believe that these revisions provide a more accurate and transparent representation of the present study and better align the manuscript with the reviewer's constructive suggestions.

We acknowledge that the present study focuses on a limited set of enzymes that are relatively well studied and, in some cases, commercially relevant. Compared to enzymes that are newly discovered, derived from native sources, or exhibit highly specialized catalytic functions, these enzymes are considered more robust in an application context. However, many studies have shown that the choice of immobilization materials and structures can substantially influence the catalytic performance of these enzymes. Our method accelerates the finding process of suitable immobilization materials. The strength of the workflow is its ability to efficiently navigate high-dimensional hybrid design spaces, reduce empirical trial-and-error, and identify favorable parameter regions with substantially fewer experimental iterations. In this context, the AI workflow serves as a practical tool for accelerating optimization and improving experimental efficiency. In fact, ongoing work in our group is extending this workflow to additional enzymes and more unique examples. We hope that the present study can serve as a methodological starting point and inspire broader application of this approach to a wider range of enzymes.

Revisions:

Manuscript, Page 20:

While the present study focused on a limited set of highly active and stable enzyme systems and employs artificial probes to transduce catalytic performance, it served as an initial demonstration of how data-driven optimization can be integrated into enzyme immobilization design. The favorable design regions identified here could, in principle, also be discovered through conventional chemical or biochemical exploration, which on the other hand typically relied on extensive trial-and-error and became increasingly inefficient as the parameter space expands. Therefore, the primary role of the PHBO-guided workflow is to accelerate the navigation of complex, high-dimensional design spaces with substantially reduced experimental effort. Future studies may define different properties, such as long-term stability or tolerance to harsh environments, enabling the algorithm to discover carriers tailored to specific operational conditions and application needs. Overall, the algorithm's adaptability to diverse materials, reaction spaces, and optimization parameters suggest its utility as an efficient method for developing high-performance biocatalytic composites.